# Differential recognition of canonical NF-κB dimers by Importin α3

Tyler J. Florio[1], Ravi K. Lokareddy[1], Daniel P. Yeggoni[1], Rajeshwer S. Sankhala[2], Connor A. Ott [1], Richard E. Gillilan [3] & Gino Cingolani [1✉]

Nuclear translocation of the p50/p65 heterodimer is essential for NF-κB signaling. In unstimulated cells, p50/p65 is retained by the inhibitor IκBα in the cytoplasm that masks the p65-nuclear localization sequence (NLS). Upon activation, p50/p65 is translocated into the nucleus by the adapter importin α3 and the receptor importin β. Here, we describe a bipartite NLS in p50/p65, analogous to nucleoplasmin NLS but exposed in trans. Importin α3 accommodates the p50- and p65-NLSs at the major and minor NLS-binding pockets, respectively. The p50-NLS is the predominant binding determinant, while the p65-NLS induces a conformational change in the Armadillo 7 of importin α3 that stabilizes a helical conformation of the p65-NLS. Neither conformational change was observed for importin α1, which makes fewer bonds with the p50/p65 NLSs, explaining the preference for α3. We propose that importin α3 discriminates between the transcriptionally active p50/p65 heterodimer and p50/p50 and p65/65 homodimers, ensuring fidelity in NF-κB signaling.

---

[1] Department of Biochemistry and Molecular Biology, Thomas Jefferson University, 1020 Locust Street, Philadelphia, PA 19107, USA. [2] Center of Infectious Disease Research, Walter Reed Army Institute of Research, Silver Spring, MD, USA. [3] Macromolecular Diffraction Facility, Cornell High Energy Synchrotron Source (MacCHESS), Cornell University, 161 Synchrotron Drive, Ithaca, NY 14853, USA. ✉email: gino.cingolani@jefferson.edu

The transport of macromolecules into and out of the cell nucleus is essential for cell physiology and is increasingly linked to the degenerative processes that lead to human diseases. The nuclear pore complex (NPC), the only gateway connecting the cytoplasm and nucleoplasm, functions as a semi-permeable barrier. While smaller molecules (<40 kDa) can diffuse through the NPC passively, most macromolecules in a cell, regardless of their size, shuttle through the NPC via an energy-dependent active transport[1–3]. Cytoplasmic cargos targeted for active nuclear import expose a nuclear localization sequence (NLS) recognized by soluble transport receptors. Several pathways for nuclear import have been described, which share similar principles and rely on soluble transport factors of the importin β superfamily (or β-karyopherins)[4]. These macromolecules promote active, signal-mediated nuclear translocation of cargos by coordinating three activities: high-affinity binding to the NLS-cargo; high-avidity association with the phenylalanine-glycine (FG) repeats lining the inner part of the NPC[5]; and, in the nucleus, binding to the small GTPase RanGTP, which promotes dissociation from FG-nups and NLS-cargo release.

Importin β can import NLS-cargos directly by either binding to their exposed NLSs[6] or through specialized adaptor proteins such as snurportin and importin α[7]. In the classical pathway, importin α binds to importin β through its N-terminal Importin-β binding (IBB) domain[8,9] and to NLS-cargos through its C-terminal 10 stacked Armadillo motifs (Arms) that collectively form an Arm-core[10,11]. Importin α possesses two NLS binding pockets located along the concave Arm-core surface: an N-terminal 'major site' and C-terminal 'minor site' between Arms 2–4 and 6–8, respectively[10,11]. There are seven genes for importin α in the human genome, classified into three clades, with importin α1 presumably functioning as the generic adapter for NLS cargos[7]. Importin α1, and to some extent other isoforms[7], is auto-inhibited by the IBB domain that folds back on itself, occupying both NLS sites[11]. In the presence of importin β and NLS-cargo, the IBB domain is displaced, and the NLS binding sites become accessible for NLS cargo recognition[8,11]. A plethora of crystal structures has elucidated the recognition of NLS peptides by importin α[12]. The classical monopartite NLS, exemplified by the SV40 T-large antigen NLS, contains a short stretch of basic residues (KKKRK) that binds primarily to the major NLS binding site and, to a lesser extent, the minor site[10]. The bipartite nucleoplasmin NLS (KR-X$_{10}$-KKKK)[12], on the other hand, contains two clusters of basic residues separated by 10–12 residues, which span both NLS-binding sites. Also, several monopartite NLSs bind selectively to either the major[13,14] or minor NLS site[8,15–17], and their interaction can be regulated by phosphorylation[18]. Recent work has revealed that importin α isoforms recognize certain NLS-cargos with a great deal of specificity. For instance, importin α3 selectively imports NF-κB p50/p65[19,20] and RCC1[21]. Likewise, viruses infecting human cells hijack specific isoforms, as shown for HIV-1 INV[22,23], influenza PB2[7], and Nipah and Hendra virus W protein[24] that are specific for importin α3 while influenza NP[25] and STAT1[26,27] appear to use importin α5. Overall, selective recognition of NLS-cargos by importin α isoforms is likely to play a role in cellular differentiation[28,29] and disease states, including viral infections, cancer[7], and developmental defects[28,30].

The nuclear factor-kappa B (NF-κB) family consists of essential transcription factors mediating various cellular processes, including immune response, cellular differentiation, and apoptosis[31–33]. There are five known NF-κB subunits; p65 (RelA), RelB, cRel, and the precursors p105 and p100, which are partially digested to form the p50 and p52 subunits, respectively[31]. These NF-κB subunits exist as a variety of homodimers and hetero-dimers, of which p50/p65 represents the most abundant and

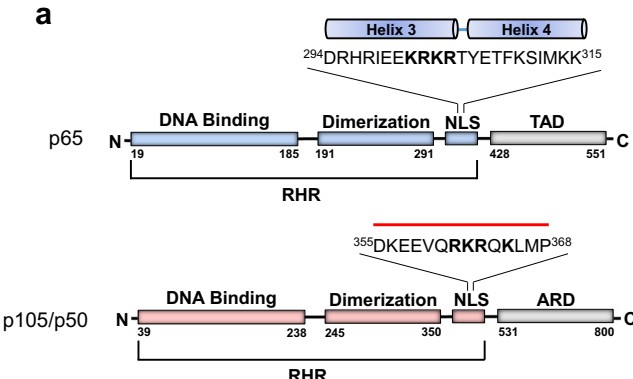

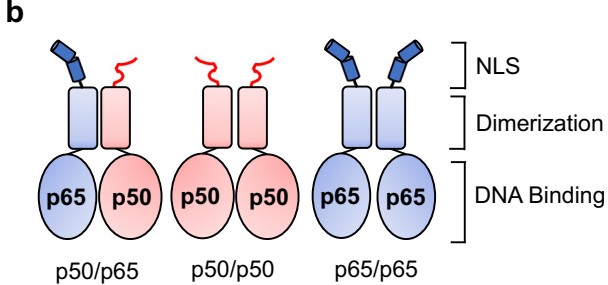

**Fig. 1 Topology of NF-κB p65 and p50. a** A schematic diagram of the p65 and p50 subunits with the position and sequence of the functional NLSs highlighted. The α-helices in the p65-NLS are based on the crystal structure of p65 bound to IκBα (PDB: 1IKN). **b** Cartoon of the NF-κB complexes with the p50- and p65-NLSs colored in red and blue, respectively.

signaling active species[31]. Overall, all NF-κB subunits contain an N-terminal Rel homology region (RHR) responsible for DNA binding, dimerization, and nuclear localization (Fig. 1a, Supplementary Fig. 1a). In contrast, only the p65, RelB, and cRel contain a C-terminal transactivation domain (TAD) responsible for gene expression[31]. The NF-κB complexes can signal through either the canonical or non-canonical (alternative) pathway, and specific NF-κB subunits regulate each pathway. In the canonical NF-κB signaling pathway, a diverse subset of stimuli, including inflammatory cytokines and chemokines, activates the p65, p50, and cRel subunits and triggers rapid and transient NF-κB response[34]. The non-canonical NF-κB pathway, on the other hand, is triggered by a smaller subset of stimuli, which activates RelB and p52 and leads to long-term, persistent NF-κB signaling[35]. The p50/p65 heterodimer, the most abundant signaling complex in the canonical NF-κB pathway, is expressed in all cell types, while the p65/p65 and p50/p50 homodimers are less abundant[31,36]. The p50/p65 cellular abundance reflects the greater structural stability of the NF-κB heterodimer compared to homodimers[36]. The p50/p65 dimerization interface is stabilized by a network of hydrogen bonds and salt bridges, including an energetically favorable interaction between Asp254 from p50 and the equivalently positioned Asn200 in p65[37,38], which is missing in the p50/p50 and p65/p65 homodimers. Comparing homodimers stability, Tyr267 and Phe307 in p50 are replaced by Phe213 and Val248 in p65, respectively, which reduce both the H-bonding and hydrophobic interactions stabilizing the p65 homodimer[36,39,40]. Thus, the NF-κB p50/p65 is more stable than p50/p50, and the p65/p65 homodimer is the least stable of the three NF-κB dimers.

The nuclear translocation of NF-κB species is tightly regulated in human cells. Fagerlund et al.[19,20] identified a functional NLS

responsible for nuclear import at the C-termini of p50 and p65 RHR (Fig. 1a, b). In the canonical NF-κB signaling pathway, the NF-κB p50/p65 and p65/p65 complexes are regulated by binding to the Inhibitor of NF-κB (IκB) proteins, which retain the complex in the cytoplasm and mask the p65-NLS[41,42]. Meanwhile, the p50 precursor p105, which lacks a TAD domain, possesses a C-terminal Ankyrin-like repeating domain (ARD) which mimics IκB and retains the precursor p105/p50 complex in the cytoplasm[43]. In the resting state, the NF-κB complex is sequestered in the cytoplasm either by IκBα-mediated retention for the p50/p65 and p65/p65 complexes[44] or ARD-mediated retention for the precursor p105/p50 complex[45,46]. Upon stimulation, the activated inhibitor of NF-κB kinase complex (NEMO/IKKβ/IKKα)[31,47] promotes IκB phosphorylation, ubiquitination, and the eventual dissociation and proteasomal degradation[48,49], which removes the IκB repression from p50/p65 and p65/p65 complexes. The newly liberated NF-κB p50/p65 complex is then shuttled into the nucleus by the isoforms importin α3 and α4, which are 95% identical in sequence[19,20]. The p65/p65 homodimer can also localize to the nucleus by binding to importin α3 and α4 as well as other isoforms, including importin α1 and importin 8[19,20,50,51]. Similarly, for the p105/p50 complex, the activated IKK-mediated signaling leads to ARD cleavage by the 20S proteasome and liberation of the p50/p50 homodimer that exposes the NLSs for importin α3/α4-mediated nuclear import[19,20,52]. Overall, the exact mechanisms of NF-κB homodimers (p50/p50 and p65/p65) nuclear import have not been elucidated. In vitro transfection studies of p65 and p50[19,20] could not fully distinguish between the heterodimeric p50/p65, the most abundant NF-κB species in human cells, and the less abundant homodimeric complexes formed by p50/p50 and p65/p65.

Here, we have investigated how NF-κB complexes interact and discriminate between importin α isoforms. Using hybrid structural methods, we found that the p50-NLS and the p65–NLS bind different regions of importin α3, generating a trans bipartite NLS. Unexpectedly, the two NLSs reveal surprising structural polymorphisms in the context of different NF-κB dimers.

## Results

**Stoichiometry of the canonical NF-κB p50/p65 nuclear import complex.** To shed light on the NF-κB nuclear import complexes' composition and structure, we purified NF-κB subunits p50 and p65 RHR, including the C-terminal NLS. Removing the TAD does not affect dimerization that is solely dictated by the dimerization domain[37,38,41,42]. We assembled and purified three NF-κB dimers (Fig. 1b, Supplementary Fig. 1b) and human importin α1 and α3 lacking the autoinhibitory N-terminal IBB domain[11] (ΔIBB-importin α1 and α3). To elucidate the association of either importin α isoform with p50/p65, we added an increasing molar excess of purified p50/p65 to 10 μg of purified ΔIBB-importin α1 (Fig. 2a) or α3 (Fig. 2b) and analyzed the resulting mixture by native gel electrophoresis. We found that only one equivalent of p50/p65 heterodimer was sufficient to entirely shift the migration of either importin α isoform, suggesting a 1:1:1 stoichiometry between ΔIBB-importin α, p50, and p65 (red star in Fig. 2a, b). We confirmed this stoichiometry using a p50/p65 heterodimer that contains a shorter version of p50 lacking the DNA-binding domain ΔDBD-p50), also competent for heterodimerization with p65[41]. The smaller ΔDBD-p50/p65 heterodimer bound one equivalent of ΔIBB-importin α3, eluting as a monodisperse 1:1:1 species by SEC (Fig. 2c). This complex was comparatively larger than ΔDBD-p50/p65 complex but similar to ΔIBB-importin α3 alone, which is elongated and can dimerize in solution[53]. The eluted complex contained three stoichiometric bands for p65, ΔDBD-p50, and ΔIBB-importin α3

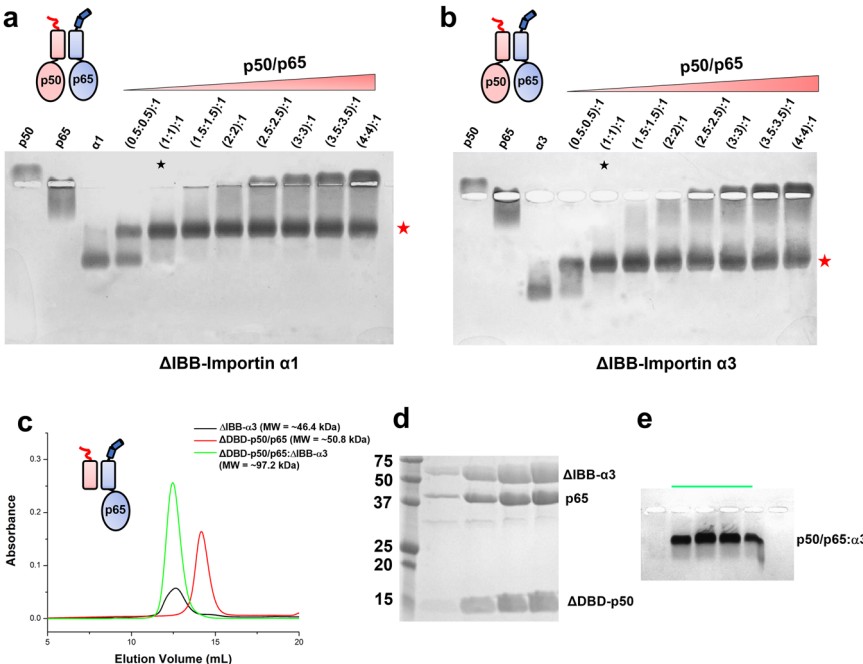

**Fig. 2 Stoichiometric binding of importin α3 to the NF-κB p50/p65 heterodimer.** Native agarose gel electrophoresis on agarose showing a titration of purified p50/p65 complex against a fixed 10 μg (~108 pmoles) of ΔIBB-importin α1 (**a**) and α3 (**b**). The p50/p65:ΔIBB-importin titration is labeled as an (X:X):1 molar ratio where (X:X) corresponds to the moles of p50 and p65 subunits. The red star denotes the p50/p65:ΔIBB-importin α complex in a stoichiometric 1:1:1 ratio. Data are representative of two independent experiments. Source data are provided as a Source Data file. **c** SEC assembly of the NF-κB trimeric p50/p65-ΔDBD:ΔIBB-importin α3 import complex (green line) from purified ΔIBB-importin α3 (gray line) and ΔDBD-p50/p65 complex (red line). The peak fraction from panel **c** was analyzed by SDS–PAGE (**d**) and on a native agarose gel (**e**). The data shown in panels **d** and **e** are representative of two independent experiments. Source data are provided as a Source Data file.

when analyzed by SDS–PAGE (Fig. 2d), but only one slowly moving species by native gel electrophoresis (Fig. 2e), indicative of a stable molecular complex. Thus, both importin α1 and α3 can bind the p50/p65 in vitro with a 1:1:1 stoichiometry, although isoform α3 is the preferred isoform in a cell[19,20].

**The p50-NLS in p50/p65 is the dominant binding determinant for importin α3**. To determine how the p65- and p50-NLSs mediate binding to importin α, we generated constructs of p65 and p50 lacking the C-terminal NLS and assembled all possible permutations of dimeric NF-κB complexes (Supplementary Fig. 1b). Using an ELISA-based microtiter binding assay, we systematically compared the binding of ΔIBB-importin α3 for NF-κB dimers containing two NLSs (p50/p65; p65/p65; p50/p50), only one NLS (ΔNLS-p50/p65, ΔNLS-p50/p65) or lacking the NLSs (ΔNLS-p50/ΔNLS-p65, ΔNLS-p65/ΔNLS-p65, ΔNLS-p50/ΔNLS-p50) (Fig. 3). Given the number of different NF-κB constructs to analyze (Supplementary Fig. 1b), we performed two sets of experiments. In the first set, we compared the binding of ΔIBB-importin α3 to four permutations of the heterodimeric p50/p65 (Fig. 3a). We found that ΔIBB-importin α3 bound the p50/p65 heterodimer with an equilibrium binding constant of $K_D = 45.8 \pm 0.9$ nM. Deleting the p50-NLS abolished association with ΔIBB-importin α3, while the p50/ΔNLS-p65 was just as potent as the wild-type NF-κB ($K_D = 59.6 \pm 2.2$ nM) though had a slightly decreased number of binding sites at equilibrium (Bmax).

In the second set of experiments (Fig. 3b), ΔIBB-importin α3 was titrated against plates pre-activated with p50/p50 or p65/p65 homodimers. This assay revealed that ΔIBB-importin α3 bound p50/p50 and p65/p65 with $K_D = 15.4 \pm 0.2$ and $35.2 \pm 0.9$ nM, respectively. Interestingly, the p50/p50 homodimer had a Bmax

twice as large as the p65/p65, suggesting a double number of ΔIBB-importin α3 bound the p50/p50 at equilibrium compared to p65/p65. Removing the NLS from either homodimer completely obliterated binding to ΔIBB-importin α3, suggesting the p65- and p50-NLSs are the major binding determinants in the context of NF-κB homodimers. To confirm the nanomolar affinities observed in vitro, both p50 and p65 dimerization domains were co-expressed in bacteria with ΔIBB-importin α3 and found to assemble into stable complexes that could be readily isolated by SEC (Supplementary Fig. 2a, b). Thus, the p50-NLS appears to be the dominant NLS in the context of the NF-κB p50/p65 heterodimer, while individual p50- and p65-NLSs are sufficient for high-affinity binding of NF-κB homodimers to importin α3.

**Competition between IκBα and importin α3**. The inhibitor IκBα binds a region of NF-κB encompassing the p65-NLS but does not contact the p50-NLS that remains solvent-exposed in the crystal structure[41,42]. Next, we explored the competition between IκBα and importin α3 for binding to NF-κB p50/p65. We purified recombinant IκBα and titrated the purified protein against a pre-formed NF-κB p50/p65 heterodimer. The resulting complex was visualized using a native agarose gel (Fig. 4a). In agreement with the literature[41,42], IκBα bound the p50/p65 stoichiometrically in a 1:1:1 molar ratio (Fig. 4a, lane 5), corresponding to one equivalent of IκBα per p50/p65 heterodimer. Excess IκBα was readily visible at the bottom of the gel, migrating as a band with faster mobility (Fig. 4a, lane 6–10). Next, we titrated increasing IκBα against a pre-formed p50/p65:α3 complex formed using a 1:1:1 ratio of the three proteins (see the control in Fig. 4a, lane 3). Even at the lowest IκBα concentration (Fig. 4a, lane 11), all importin α3 was

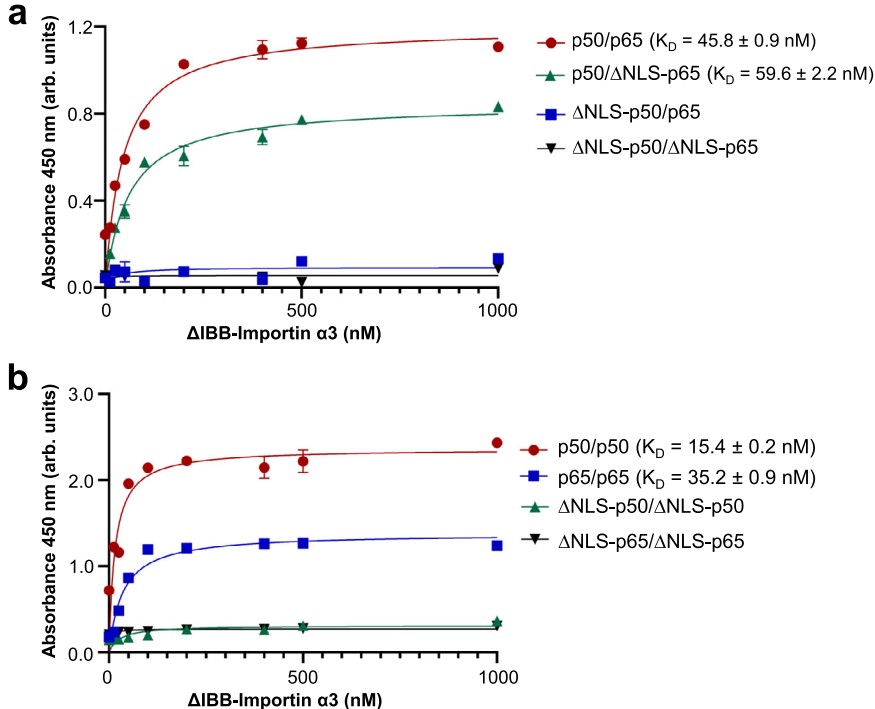

**Fig. 3 ELISA-based microtiter binding assays. a** ΔIBB-importin α3 was titrated against p50/p65 (red), p50/ΔNLS-p65 (green), p50-ΔNLS/p65 (blue), ΔNLS-p50/ΔNLS-p65 (black). The dissociation constant ($K_D$) of ΔIBB-importin α3 for p50/p65 and ΔNLS-p50/ΔNLS-p65 is $K_D = 45.8 \pm 0.9$ and $59.6 \pm 2.2$ nM, respectively. Non-saturable binding was observed for ΔNLS-p50/p65 and ΔNLS-p50/ΔNLS-p65. **b** ΔIBB-importin α3 was titrated against p50/p50 (red), p65/p65 (blue), ΔNLS-p50/ΔNLS-p50 (green), and ΔNLS-p65/ΔNLS-p65 (black). The dissociation constant ($K_D$) of ΔIBB-importin α3 for p50/p50 and p65/p65 is $K_D = 15.4 + 0.2$ and $35.2 + 0.9$ nM, respectively. No measurable binding was observed for ΔNLS-p65/ΔNLS-p65 and ΔNLS-p50/ΔNLS-p50. The standard deviation was calculated from averaging two independent experiments. Source data are provided as a Source Data file.

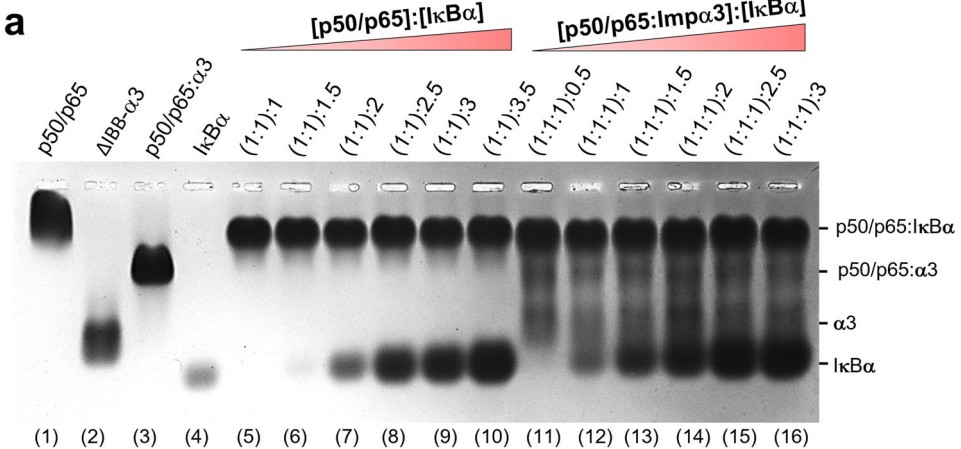

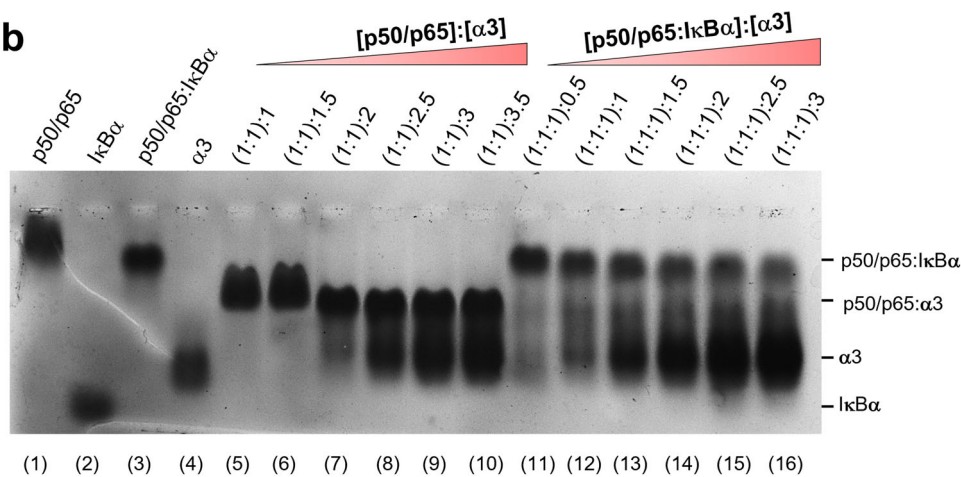

**Fig. 4 Competition between IκBα and ΔIBB-importin α3 for NF-κB p50/p65. a** Native agarose gel electrophoresis showing a titration of purified IκBα against ~130 pmoles of pre-formed p50/p65 (lanes 5–10) or p50/p65:ΔIBB-importin α3 complex (lanes 11–16). The moles of IκBα are relative to those of individual species in pre-formed dimeric and trimeric complexes. All controls in lanes 1–4. **b** Native agarose gel electrophoresis showing a titration of purified ΔIBB-importin α3 against ~130 pmoles of pre-formed p50/p65 (lanes 5–10) or p50/p65:IκBα (lanes 11–16). The moles of ΔIBB-importin α3 are relative to those of individual species in pre-formed dimeric and trimeric complexes. The data shown in panels **a** and **b** are representative of two independent experiments. Source data are provided as a Source Data file.

displaced from p50/p65, giving rise to the p50/p65:IκBα complex (Fig. 4a, lanes 11–16). We also performed a converse experiment whereby a stoichiometric p50/p65:IκBα complex was first formed using a 1:1:1 molar ratio of p50/p65 and IκBα (Fig. 4b, lane 3), and then challenged with an increasing molar excess of purified importin α3 (Fig. 4b, lane 11–16). Even at a super-physiological importin α3 concentration of ~54 μM, over 50× higher than the estimated cellular concentration for all isoforms[54], the inhibited p50/p65:IκBα complex remained intact, and there was no appearance of the p50/p65:α3 complex on the gel (shown in Fig. 4b, lane 5). Overall, these data suggest that IκBα and importin α3 binding to p50/p65 are mutually exclusive. Though exposed in the trimeric p50/p65:IκBα complex[41,42], the p50-NLS is not accessible for association with importin α3 when IκBα is present.

**Crystallographic analysis of importin α isoforms 1 and 3 bound to NF-κB NLSs.** To visualize NF-κB's association with importins, we attempted crystallization of the p50/p65 heterodimer in complex with importin α isoforms but were unsuccessful. Large crystals of the NF-κB p50/p50 homodimer were obtained with ΔIBB-importin α3 but failed to diffract X-ray

beyond 10 Å resolution. We then turned to study the structure of importin α isoforms 1 and 3 in complex with p50- and p65-NLS peptides (Fig. 1a). In total, we obtained and solved five crystal structures (Table 1): two complexes of human ΔIBB-importin α3, either bound to the p50-NLS alone or both p65- and p50-NLS peptides (Fig. 5a, b), and three complexes of the mammalian ΔIBB-importin α1 bound to p65-NLS or p50-NLS alone, as well as both p65- and p50-NLS peptides (Fig. 5d–f). The best importin α3 crystals diffracted to 2.10 Å for the p50-NLS complex and 2.85 Å with p65/p50-NLS peptides. Similarly, the best diffraction data for importin α1 (~2.20 Å) were obtained from crystals bound to just the p50-NLS, while a complex with both the p65/p50-NLSs or just the p65-NLS diffracted to 2.40 and 2.82 Å resolution, respectively (Table 1). Molecular replacement for the NF-κB NLS/ importin α3 complexes was initially attempted using the apo importin α3 structure[24] as a search model but yielded a poor solution with a high clash score. In contrast, the Nipah virus W protein-bound importin α3 structure[24] yielded a straightforward solution, which could be easily refined to an Rfree under 30%. All crystal structures had clear electron densities for the p50- and p65-NLSs and adjacent flanking regions that we modeled in unbiased $F_0 - F_c$ electron density maps and confirmed in Polder

**Table 1 Crystallographic data collection and refinement statistics.**

| | ΔIBB-Imp α3 + p50-NLS | ΔIBB-Imp α3 + p50-NLS + p65-NLS | ΔIBB-Imp α1 + p65-NLS | ΔIBB-Imp α1 + p50-NLS | ΔIBB-Imp α1 + p50-NLS + p65-NLS |
|---|---|---|---|---|---|
| *Data collection* | | | | | |
| Beamline | SSRL 12-1 | SSRL 12-1 | SSRL 9-2 | SSRL 12-1 | SSRL 12-1 |
| Wavelength (Å) | 0.953 | 0.979 | 0.979 | 0.979 | 0.979 |
| Space group | $P2_1$ | $P3_221$ | $P2_12_12_1$ | $P2_12_12_1$ | $P2_12_12_1$ |
| Cell dimensions *a, b, c* (Å) | 47.6, 59.0, 86.0 | 117.1, 117.1, 210.2 | 78.5, 90.8, 99.8 | 78.1, 90.7, 97.4 | 77.9, 91.2, 97.2 |
| $\alpha, \beta, \gamma$ (°) | 90.0, 96.3, 90.0 | 90.0, 90.0, 120.0 | 90.0, 90.0, 90.0 | 90.0, 90.0, 90.0 | 90.0, 90.0, 90.0 |
| Reflections (tot/unique) | 1,021,555/27,642 | 471,279/34,874 | 665,708/17,271 | 768,008/27,626 | 370,915/24,702 |
| Resolution (Å) | 30.0–2.10 (2.16–2.10) | 50.0–2.82 (2.92–2.82) | 15.0–2.81 (2.91–2.81) | 15.0–2.24 (2.33–2.24) | 50.0–2.40 (2.49–2.40) |
| Completeness (%) | 98.8 (87.7) | 85.4 (45.5) | 97.7 (89.3) | 81.9 (87.1) | 89.2 (57.8) |
| Redundancy | 5.5 (4.3) | 3.1 (2.9) | 6.9 (4.6) | 3.8 (3.0) | 3.1 (2.2) |
| $R_{sym}$ | 6.4 (55.2) | 13.7 (85.3) | 11.1 (53.0) | 9.1 (85.2) | 12.4 (81.6) |
| $R_{pim}$ | 4.4 (41.2) | 8.2 (71.2) | 7.3 (73.0) | 7.7 (74.8) | 8.0 (70.8) |
| $I/\sigma_I$ | 11.6 (2.4) | 6.3 (1.2) | 14.5 (1.5) | 14.4 (1.5) | 13.2 (1.2) |
| CC1/2 | 0.998 (0.815) | 0.994 (0.287) | 0.995 (0.718) | 0.996 (0.309) | 0.988 (0.409) |
| Wilson *B*-factor (Å$^2$) | 42.0 | 64.1 | 70.1 | 38.2 | 51.6 |
| *Refinement* | | | | | |
| PDB entry | 7LFC | 7LF4 | 7LEU | 7LEQ | 7LET |
| Resolution limits (Å) | 15.0–2.10 | 15.0–2.85 | 15–2.82 | 15–2.24 | 50–2.40 |
| No. of reflections | 27,534 | 25,877[a] | 17,116 | 27,564 | 24,529 |
| $R_{work}/R_{free}$[b] | 19.9/22.0 | 19.6/23.7 | 20.0/23.5 | 19.1/21.8 | 19.8/24.1 |
| No. of copies Asym. Unit | 1 | 2 | 1 | 1 | 1 |
| No. of protein atoms | 3302 | 6859 | 3361 | 3291 | 3358 |
| No. of solvent | 105 | 0 | 1 | 149 | 32 |
| Ramachandran (%) allow/gener/ disallowed Rms from ideal | 98.34/1.66/0.0 | 96.20/3.80/0.0 | 97.22/2.78/0.0 | 98.12/1.88/0.0 | 97.44/2.56/0.0 |
| Bond lengths (Å) | 0.009 | 0.005 | 0.007 | 0.006 | 0.006 |
| Bond angles (°) | 1.06 | 1.08 | 1.05 | 1.09 | 0.82 |
| MolProbity Score | 1.38 | 1.87 | 1.52 | 1.45 | 1.41 |
| MolProbity Clashscore | 6.33 | 12.6 | 6.76 | 8.39 | 6.17 |

Values in parentheses are for the highest-resolution shell.
[a]Reflections with |Fobs |/σFobs > 2.15.
[b]The Rfree value was calculated using 5% of randomly selected reflections.

maps (Supplementary Figs. 3 and 4). The final models have Rfree under 24% and excellent stereochemistry (Table 1), allowing for a direct structural comparison of the five crystallographic structures.

**The p50-NLS binds exclusively to the major NLS-binding pocket.** Both ΔIBB-importin α3 and α1 crystallized with an excess of the p50-NLS had strong electron density for the NLS only at the major NLS binding site (Supplementary Figs. 3a and 4a). The two isoforms adopt a nearly identical conformation of the Arm core, and the p50-NLS $P_1$–$P_5$ residues (Arg361-Lys365) are not only superimposable in the two complexes (RMSD of 1.095) (Fig. 6a) but also occupy equivalent positions. However, the p50-NLS binds more intimately with importin α3 than α1, contributing 18 hydrogen bonds, two salt bridges, 152 Van der Waals contacts, and burying a surface area of 855.8 Å$^2$ (Fig. 6c) versus nine hydrogen bonds, no salt bridges, 118 Van der Waals contacts, and burying a surface area of only 780.4 Å$^2$ in α1 (Fig. 6e). Accordingly, eleven residues of the p50-NLS (res. 358–368) were modeled in the importin α3 structure versus only seven (res. 361–367) for importin α1. In both complexes, the p50-NLS residues [361]RKRQK[365] occupy the $P_1$–$P_5$ positions at the major NLS-binding pocket (Supplementary Table 2), with Lys362 at the crucial $P_2$ position. Additionally, the p50-NLS contains an N-terminal extension [356]KEEVQ[360] immediately upstream of the NLS that bears a considerable amount of negatively charged residues, including Glu357, which makes a salt bridge with importin α3 Arg306 in the importin α3 + p50-NLS complex

(Fig. 6c). Comparing the p50-NLS sequence to the NLSs of hPLSCR1, Ku70, and Ku80, which also binds only at the major NLS site[13,14], reveals conservation of a Lys at the $P_5$ site (Supplementary Table 2) that makes hydrogen bonds with Gln176 and Asn141, possibly explaining why the p50-NLS targets the major NLS-binding pocket only. Thus, the crystallographic analysis of importin α1 and α3 bound to the p50-NLS reveals this NLS binds exclusively to the major NLS-binding pocket, explaining the specificity for the isoform α3 reported in the literature[19].

**The p65-NLS binds differently to the major and minor NLS-pockets.** Only the isoform α1 crystallized bound to the p65-NLS alone. A 2.8 Å crystal structure revealed two p65-NLS peptides bound to the major and minor NLS pockets (Fig. 5e, Supplementary Fig. 4b) in slightly different conformations. At the major site (Fig. 6g), the p65-NLS occupies sites $P_1$–$P_5$ in a p50-NLS-like conformation, with Glu300, Lys301, Arg302, Lys303, and Arg304, making a total of 11 hydrogen bonds, 101 Van der Waals contacts, and burying 724.2 Å. Unlike the p50-NLS, which inserts Arg361 into the $P_1$ position and is stabilized by Van der Waals interactions, the Glu300 in the p65-NLS orients away from the $P_1$ position (Fig. 6g). The p65-NLS also occupies the minor NLS pocket (Fig. 6h). This binding includes ten hydrogen bonds and 88 Van der Wal contacts but no salt bridges, with the Arg302 occupying the $P_2'$ position (Supplementary Table 2).

Interestingly, adding both p65- and p50-NLS peptides to importin α3 (Fig. 5b) and α1 (Fig. 5f) resulted in strong electron density for the p65-NLS peptide at the minor site, with the

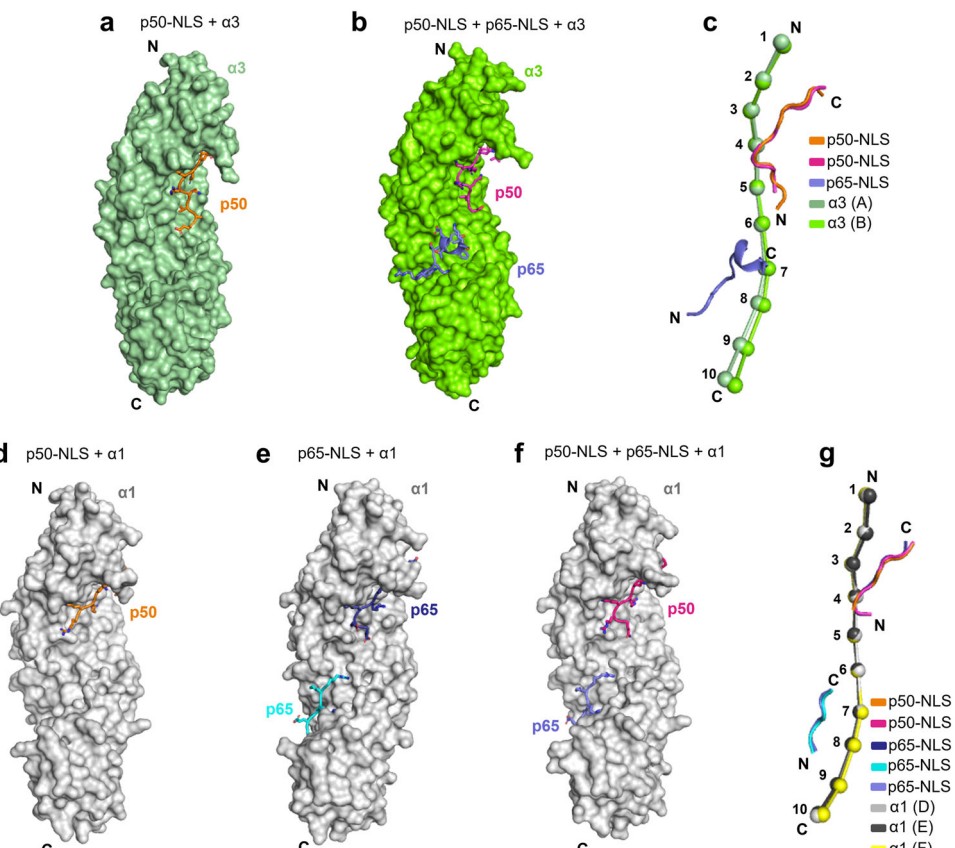

**Fig. 5 Crystal structures of ΔIBB-importin α3 and α1 bound to NF-κB p65- and p50-NLS peptides.** Surface representation of ΔIBB-importin α3 (colored in dark and light green) bound to the (**a**) p50-NLS (orange) (**b**) p50- and p65-NLSs (magenta and blue), respectively. ΔIBB-importin α3 bound to p50- and p65-NLSs crystallized as a dimer, but only one chain (chain C) is shown in panels **b**, **c**. The two chains in the asymmetric unit are nearly identical (RMSD = 1.14 Å), but the dimer is likely a crystallographic artifact due to the high concentration used for crystallization. Both importin α3 molecules in the asymmetric unit are bound to the p50- and p65-NLS peptides, which occupy identical binding sites and have similar structures. However, one of the importin α3 molecules (chain A) has a higher B-factor, likely due to weaker crystal contacts, which prompted us to use chain C for structural analysis. **c** Superimposition of importin α3 structures in A (dark green) and B (light green) shown as beads-on-a-string. Surface representation of ΔIBB-importin α1 (colored in light gray) bound to the (**d**) p50-NLS (orange), (**e**) p65-NLS (dark blue and cyan), and (**f**) p50- and p65-NLSs (magenta and blue). **g** Superimposition of all importin α1 structures in **d** (light gray), **e** (dark gray), and **f** (yellow) shown as beads-on-a-string.

p50-NLS firmly occupying the major NLS-site (Supplementary Figs. 3b and 4c). However, there were noticeable differences between the p65-NLS conformations and contacts made with either isoform (Fig. 6b). In the importin α3 co-crystal structure with the p65- and p50-NLSs (Fig. 6d), the eleven residue p65-NLS can be divided into two stretches, with the N-terminal residues [301]KRKRT[305] adopting an extended conformation and the C-terminal residues [306]YETFKS[311] folding into an α-helix. The residues [303]KRTYE[307] occupy the $P_1'$-$P_5'$ positions, including the helical conformation contributing to a salt bridge between p65-NLS Glu307 and importin α3 Lys231 within the Arm 4 coiled region (Supplementary Table 2). The p65-NLS helical conformation resembles the p65 helical structure visualized in the IκBα- and IκBβ-NF-κB complexes[41,42] with the helix 4 (res. 305–311) retained and bound to both IκBα and importin α3. In contrast, the p65-NLS residues 294-303 fold into α-helix (referred to as helix 3) in the IκB-bound complex but bind importin α3 in an extended conformation. Unexpectedly, in all complexes with importin α1, the p65-NLS adopts a fully extended conformation at the minor NLS-site, with the $P_1'$-$P_4'$ positions occupied by [301]KRKR[304] residues (Fig. 6f, h); the α-helix [306]YETFKS[311] seen in complex with importin α3 (Fig. 6d) is missing in these structures.

The mechanism by which different isoforms accommodate the p65-NLS at the minor NLS-binding pockets determines the number and types of contacts made in the two complexes. PISA[55] revealed the p65-NLS:importin α3 binding ($\Delta G = -0.6$ kcal/mol) is more energetically favorable than the p65-NLS:importin α1 ($\Delta G = 4.9$ kcal/mol), which possibly explains the physiological preference for importin α3. Additionally, comparing the p65-NLS in both importin α isoforms, there appears to be a shift in the NLS register at the $P_2'$ position. In the importin α1 structures (Fig. 6f, h), the $P_2'$ position is occupied by Arg302 while Lys301, Lys303, and Arg304 occupy the $P_1'$, $P_3'$, and $P_4'$, respectively, and the $P_5'$ is unoccupied. In contrast, in complex with importin α3 (Fig. 6d), the $P_2'$ position is occupied by Arg304, with Thr305, Tyr306, and Glu307 at the $P_3'$-$P_5'$ positions, respectively. The residues responsible for generating this frameshift are the same that form the α-helix ([306]YETFKS[311]) observed in the importin α3/p65–NLS complex and are part of the α-helix 4 (res. 306–321) responsible for binding to IκBα[41,42].

**A conformational change in importin α3 Arm-core upon NF-κB NLS recognition.** We compared the structures of importin α3 crystallized with different p50- and p65-NLS peptides and found importin α3 adopts a significantly different conformation when the p65-NLS is bound at the minor NLS-box (Fig. 5c). Importin α3 flexible solenoid can be rationalized as two arches formed by Arms 1–4 and 7–10 flapping dynamically around a central core

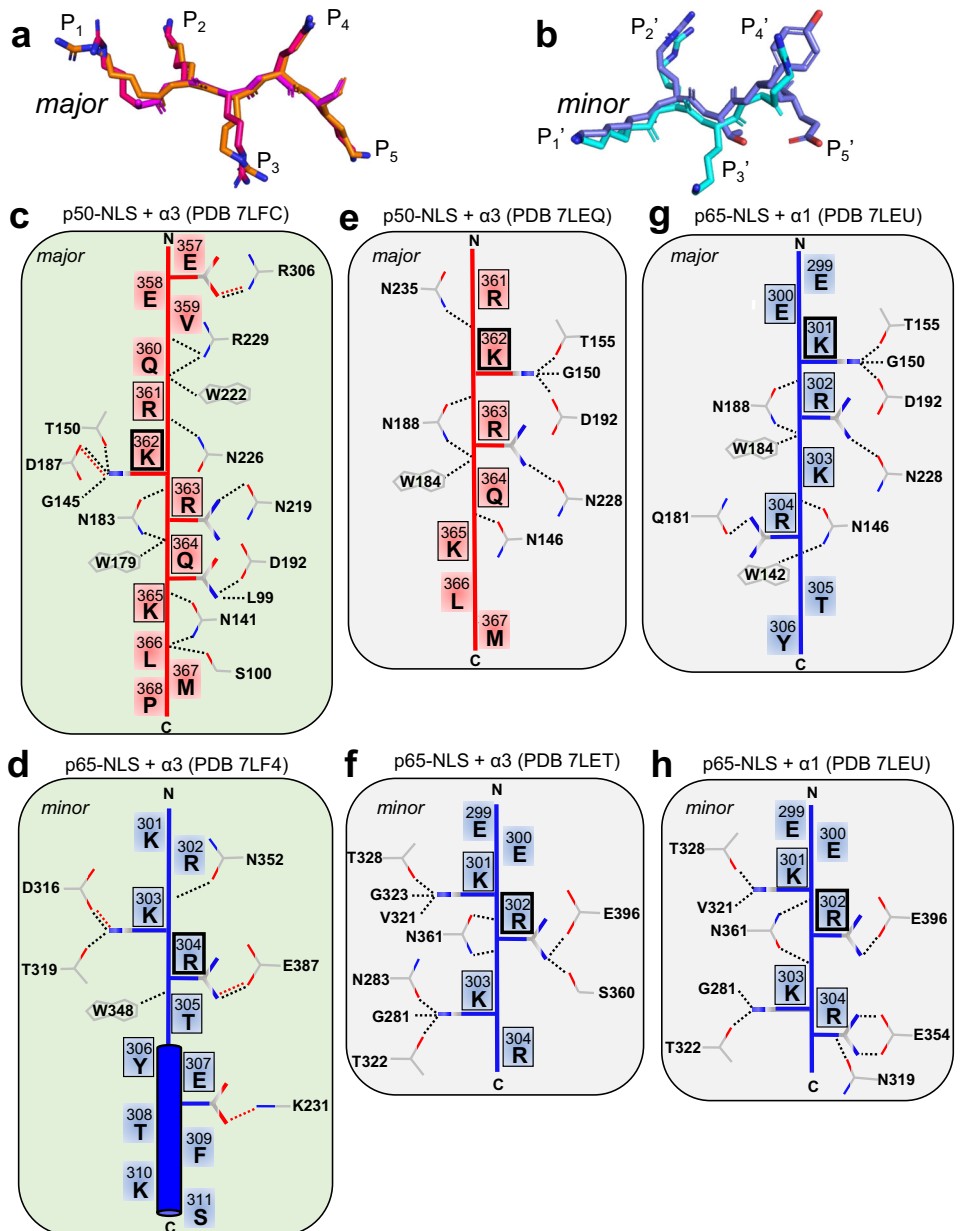

**Fig. 6 Schematic of NLS-binding at the major and minor NLS binding site of importin α isoforms. a, b** Overlay of the p50-NLSs and p65-NLSs P₁–P₅ positions bound to the major and minor NLS-binding sites of ΔIBB-importin α1 (orange and cyan, respectively) and α3 (magenta and blue, respectively). **c–h** Schematic diagram of all major contacts made by the p50-NLS (red) and p65-NLS (blue) with ΔIBB-importin α3 and α1, at the major (panels **c**, **e**, **g**) and minor (panels **d**, **f**, **h**) NLS-binding sites. All contacts shown occur within 3.5 Å distance: hydrogen bonds and salt bridge interactions are shown by dash lines colored in black and red, respectively. NLS residues occupying positions P₁–P₅ and P₁'–P₅' are marked with a black frame (thicker for positions P₂ and P₂').

consisting of Arms 5–6[8]. DynDom analysis[56] of the importin α3 crystal structures in the unliganded state and bound to p50-NLS and p65-/p50-NLSs revealed two structural changes in the peripheral arches in response to the p65-NLS binding. At the N-terminus, a hinge movement[57] was identified within Arm 4 (res. 198-201) in response to p50-NLS and p50-/p65-NLS binding compared to the apo structure (Supplementary Fig. 5a, b). This hinge movement makes the major site more accessible for the p50-NLS and facilitates numerous intimate p50-NLS contacts. Similarly, DynDom revealed a shear movement within Arm 7 residues 337–338 only in the importin α3 structure with p65-/p50-NLSs (Supplementary Fig. 5b, c). This shear movement, described as a movement along the plane of an interface[57], exposes an additional ~93 Å surface area and shifts the C-terminal Arm 10 by ~3.5 Å

compared to the apo importin α3 structure. These conformational changes facilitate the closure of the C-terminal arch around the p65-NLS, while Arms 2–4 cradles the p50-NLS peptide. Overall, the segmented motion of importin α3 Arm core allows the minor site to become more accessible, accommodating the partially helical p65-NLS, as previously reported for other α3-specific NLS-cargos[24,58].

SuperPose[59] revealed importin α3 bound to the p50- and p65-NLSs aligns well at both major and minor NLS sites with the structure of importin α3 bound to SOX2[58] (RMSD 1.54 Å between residues 72 and 486) (Supplementary Fig. 6). However, the structure of p50/p65:α3 differs significantly from the RCC1-bound importin α3[21] (RMSD 2.00 Å between residues 72–487) and even more from the apo α3[24] (RMSD 2.80 Å between

residues 72 and 486). Meanwhile, importin α1 residues 75–497 are perfectly superimposable (RMSD 0.26–0.39 Å) in all complexes of importin α1 bound to either p65-, p50-, and both NLSs (Fig. 5g), underscoring the rigid structure of this isoform[7] that can stretch NLS peptides at the major and minor binding sites by making extensive mainchain contacts. Thus, the intrinsic plasticity of importin α3 appears to play a role in NLS-cargo specificity in contrast to the more rigid importin α1.

**Solution structure of NF-κB p50/p65 bound to importin α3**. To investigate the quaternary structure of different NF-κB import complexes, we performed size-exclusion chromatography coupled with small-angle X-ray scattering (SEC–SAXS)[60]. A pre-assembled complex of NF-κB p50/p65 (Fig. 7a) bound to ΔIBB-importin α3 was analyzed on an analytical Superdex gel filtration column and eluted fractions were subjected to SAXS using a 0.25 mm × 0.25 mm microbeam. The resulting SEC–SAXS profiles were proven suitable for biophysical analysis. We found the p50/p65:ΔIBB-importin α3 complex has a radius of gyration ($R_g$) of 44.21 ± 0.28 Å (Fig. 7a, Supplementary Table 3), and a volume of correlation[61] mass of 124.4 ± 12.4 kDa, in close agreement with the theoretical MW ~ 129 kDa expected for a heterotrimer of p50/p65:ΔIBB-importin α3 in a 1:1:1 stoichiometry (Supplementary Table 1). The distance distribution function $P(r)$ calculated from the SAXS data indicates a maximum diameter $D_{max}$ ~ 137 Å (Fig. 7d). We calculated an electron density from solution scattering data using DENSS (Supplementary Table 3), which is shaped like a 'fidget spinner' with two smaller 'feet' domains and a slightly larger 'hood' domain sitting on top. We interpreted this density by docking the crystal structures the p65 and p50 DNA-binding domains solved in the presence of DNA[37] inside the feet domains, which are loose with respect to the dimerization domain in the absence of DNA. In contrast, the crystal structure of ΔIBB-importin α3 bound to p65/p50-NLSs was manually docked into the larger hood density, which is perched perpendicular to the p50/p65 heterodimer (Fig. 8a). The directionality of importin α3 with respect to the p50/p65 core was determined based on the NLS-peptide bound to the major (p50) and minor (p65) NLS-binding sites that accurately position the N- and C-termini of importin α3 close to p50 and p65, respectively. This pseudo-atomic model was refined against the DENSS density using rigid-body refinement, yielding an excellent agreement between the atomic model and SAXS-data ($\chi^2 = 1.50$) (Supplementary Fig. 7a).

We also studied the solution structure of homodimeric p50/p50 and p65/p65 bound to ΔIBB-importin α3 (Supplementary Table 3). The p50/p50:ΔIBB-importin α3 complex appeared larger in solution than the p50/p65:ΔIBB-importin α3 complex, with an $R_g$ of 48.56 + 0.22 Å (Fig. 7b) corresponding to an approximate hydrated mass of 200 ± 20 kDa. This observation is consistent with a bi-dimeric complex containing two importins α3 per p50/p50 homodimer (stoichiometry 2:2) with MW ~ 185.2 kDa (Supplementary Table 1). The distance distribution function $P(r)$ calculated from the SAXS data indicates a maximum diameter $D_{max}$ ~ 157 Å (Fig. 7d). The calculated electron density from DENSS has four lobes with two 'feet' and two 'arms' protruding upward. We interpreted the density by docking in the p50 DNA-binding domain solved without the DNA in the feet domain, the dimerization domain projecting into the center, and two crystal structures of the ΔIBB-importin α3 bound to p50-NLSs manually docked into the density (Fig. 8b). The pseudo-atomic model was refined against the DENSS density using rigid-body refinement, yielding an excellent agreement between the atomic model and SAXS-data ($\chi^2 = 1.29$) (Supplementary Fig. 7b). In contrast, the p65/p65:ΔIBB-importin α3

complex has an $R_g$ of 44.50 ± 0.24 Å (Fig. 7c, Supplementary Table 3), corresponding to an approximate hydrated mass of 146 ± 14.6 kDa, close to a 1:1:1 complex (MW ~ 125.4 kDa) (Supplementary Table 1), as observed for the p50/p65 heterodimer. The distance distribution function $P(r)$ calculated from the SAXS data indicates a maximum diameter $D_{max}$ ~ 151 Å, which is slightly larger than the p50/p65:ΔIBB-importin α3 complex (Fig. 7d). The calculated electron density from DENSS is also shaped like a fidget spinner with the importin α3 perched laterally instead of perpendicular to the dimerization domain (Fig. 8c), similar to the p50/p65:ΔIBB-importin α3. We interpreted the density by docking the p65 DNA-binding domain solved without the DNA (PDB: 1RAM) in the feet domain, the dimerization domain projecting into the center, and manually docked the crystal structures of the ΔIBB-importin α1 bound to p65-NLSs into the density. The pseudo-atomic model was refined against the DENSS density using rigid-body refinement, yielding an excellent agreement between the atomic model and SAXS-data ($\chi^2 = 1.71$) (Supplementary Fig. 7c). Interestingly, these results suggest both NF-κB complexes containing the p65-NLS binds one equivalent of importin α3, consistent with a 1:1:1 quaternary structure. Meanwhile, in the p50/p50 homodimer, the p50-NLS binds two copies of importin α3, consistent with a 2:2 quaternary structure. Thus, the NF-κB complex composition is the key determinant for asymmetric recognition by importin α3.

## Discussion

The specificity of importin α3 for the canonical NF-κB p50/p65 heterodimer was first reported over fifteen years ago[19,20]. However, the mechanisms of NF-κB nuclear import have remained poorly studied, despite the vital role of this transcription factor in cell physiology and disease. In this paper, we elucidate the structural basis governing the recognition of canonical NF-κB heterodimer and homodimers by the isoform importin α3.

**NF-κB p50/p65 subunits generate a bipartite NLS in trans**. We provide biochemical (Fig. 2) and biophysical (Fig. 7) evidence that the p50, p65, and importin α3 assembles into a 1:1:1 heterotrimeric complex, supporting a previous model[19]. The p50- and p65-NLSs are recognized asymmetrically by importin α3, which sequesters them at the major and minor NLS-pocket, respectively. This recognition is similar to that described for the dimeric transcription factor STAT1, which is also recognized asymmetrically by importin α5[26,27]. However, NF-κB is more complex than STAT1 due to the existence of different dimeric complexes in living cells. We found that all p65-containing NF-κB dimers assembled into a 1:1:1 complex with importin α3, while the p50/p50 homodimer assembled in a 2:2 complex. These results suggest the p65 subunit is key to introduce asymmetry in the NF-κB nuclear import complex. Crystallographic and SAXS studies revealed that the two NF-κB subunits generate a bipartite NLS in trans, similar to some oligomeric viral complexes[25,62] that expose multiple NLSs on their quaternary structure[25,79]. While the NF-κB p50/p65 resembles a bipartite NLS-like cargo, it does not show the classical bipartite NLS interdependence where both NLS boxes are necessary, and mutation of either cluster of basic residues significantly impacts importin α binding, at least in vitro[63]. In our binding studies (Fig. 3a), only the p50-NLS was indispensable for importin α3 association, suggesting this NLS is the primary determinant for nuclear import, in agreement with a prior transfection study[19]. Our crystal structures help rationalize this specificity by showing that p50- and p65-NLSs bind more intimately to importin α3 than importin α1. The p50-NLS occupies the major NLS pocket making 18 hydrogen bonds and

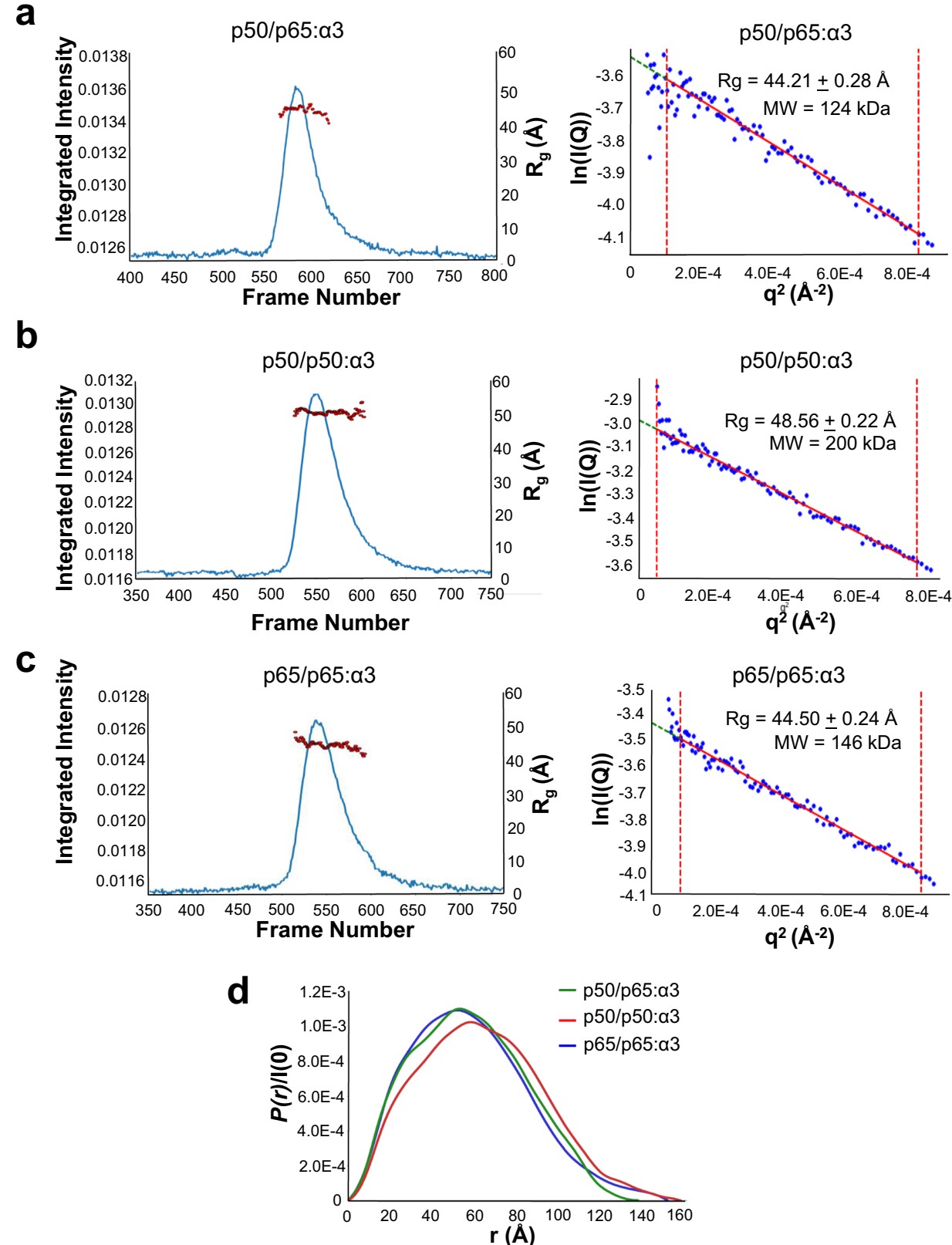

two salt bridges in complex with importin α3 versus only nine hydrogen bonds with importin α1.

Similarly, the binding of the p65-NLS to importin α3 is also energetically more favorable than importin α1. Unexpectedly, the p65-NLS adopts two strikingly different conformations bound to importin α3 or α1. It is fully extended and stretched with importin α1, both at the major and minor NLS pockets (Fig. 6f, h), while it binds to the importin α3 minor NLS pocket in an extended conformation up to the $P_3'$ position but is helical from the $P_4'$ position onward (Fig. 6d). A helical conformation of the NLS backbone has been previously reported[15,17,64], though it remains relatively rare, as most NLSs

**Fig. 7 SEC–SAXS analysis of pre-formed NF-κB/ΔIBB-importin α3 complexes. a** SEC–SAXS profile of a preassembled p50/p65/ΔIBB-importin α3 complex at 4.5 mg ml$^{-1}$ measured in 20 mM Tris–HCl pH 7.0, 0.175 M NaCl, 2.5% glycerol and 0.5 mM TCEP at 4 °C. The red dots indicate $R_g$ values (on the Y-axis) corresponding to frames on the X-axis. The right panel shows the Guinier plot calculated from averaging buffer-subtracted scattering intensities from frames (465–484) and the sample peak (571–599) (red dots). The p50/p65/ΔIBB-importin α3 coefficient of determination for a best-fit line, $R^2$, is 0.9564 (red line). **b** SEC–SAXS profile of the pre-formed p50/p50/ΔIBB-importin α3 complex at 6.29 mg ml$^{-1}$ measured in 20 mM Tris–HCl pH 7.5, 0.15 M NaCl, 2.5% glycerol and 0.5 mM TCEP at 4 °C. The red dots indicate $R_g$ values (on the Y-axis) corresponding to frames on the X-axis. On the right side is a Guinier plot calculated from averaging buffer-subtracted scattering intensities from frames (180–242) and the sample peak (547–556) (red dots). The p50/p65/ΔIBB-importin α3 coefficient of determination for a line of best fit, $R^2$, is 0.9766 (red line). **c** SEC–SAXS profile of the pre-formed p65/p65/ΔIBB-importin α3 complex at 6.60 mg ml$^{-1}$ measured in 20 mM Tris–HCl pH 7.5, 0.15 M NaCl, 2.5% glycerol and 0.5 mM TCEP at 4 °C. The red dots indicate $R_g$ values (on the Y-axis) corresponding to frames on the X-axis. The Guinier plot (right panel) was calculated from averaging buffer-subtracted scattering intensities from frames (72–118) and the sample peak (538-550) (red dots). The p65/p65/ΔIBB-importin α3 coefficient of determination for the line of best fit, $R^2$, is 0.9712 (red line). **d** $P(r)$ function plots calculated from SEC–SAXS data for p50/p65/ΔIBB-importin α3 (green), p65/p65/ΔIBB-importin α3 (blue) and p50/p50/ΔIBB-importin α3 (red) complexes.

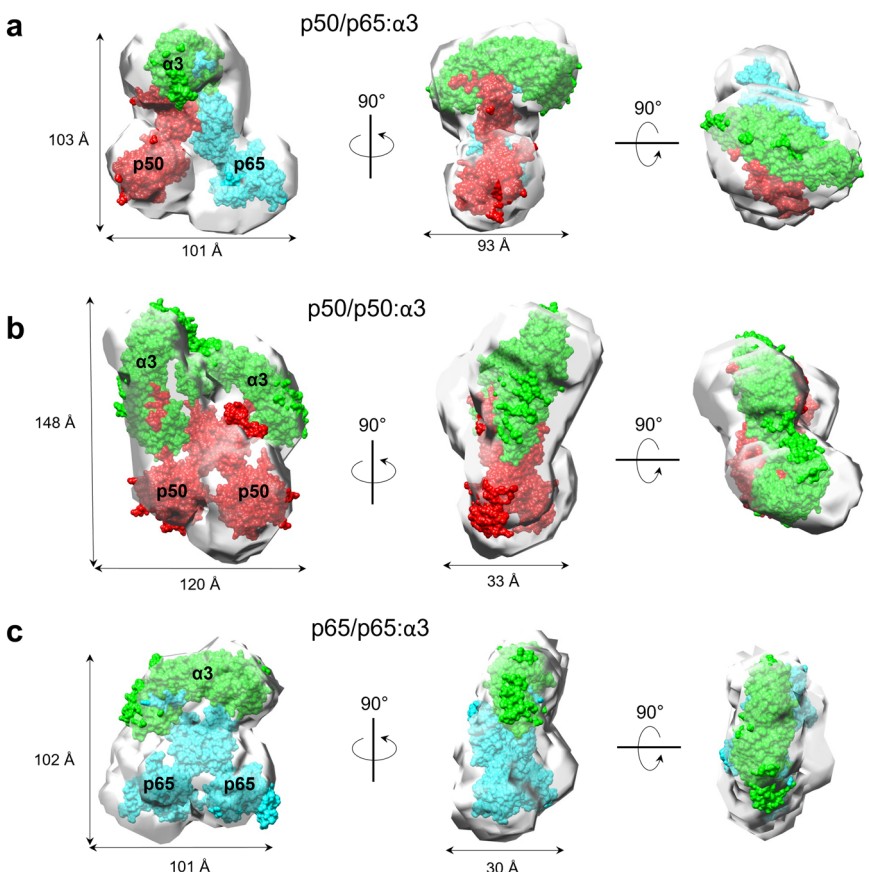

**Fig. 8 Solution structures of the NF-κB import complexes.** Ab initio SEC–SAXS electron density of **a** p50/p65/ΔIBB-importin α3, **b** p50/p50/ΔIBB-importin α3 (**c**) and p65/p65/ΔIBB-importin α3. All DENSS densities are shown as semitransparent surfaces. The crystallographic structures of ΔIBB-importin α3 (green), p50/p50 (PDB: 1NFK) (red) and p65/p65 homodimers (PDB: 1RAM) (blue), and p50/p65 heterodimer (blue/red) (PDB: 1VKX) were docked inside the density and are shown as a surface representation. The agreement between solution and modeled crystal structures was calculated using the FoXS server and shown in Supplementary Fig. 7.

bind importin α in an extended conformation to maximize binding contacts with the NLS-binding pocket. Similar to other helical NLSs[15,17,64], the C-terminal region of the p65-NLS contains aromatic residues, like Tyr306 and Phe309. NMR studies found the region of p65 encompassing the NLS to be predominantly random coil in solution, but it folds into helix 3 (res. 293–304) and helix 4 (res. 306–315) in complex with IκB[41,42,65–67]. While our data suggests the p65-NLS is not a major binding determinant for importin α3 in the p50/p65 heterodimer, it is a functional NLS for the p65/p65 homodimer (Fig. 3b). In the absence of p50, the p65-NLS binds the major NLS site, mimicking the p50-NLS (Fig. 5e). If the p65-NLS is not a major determinant

for nuclear import of p50/p65, why is it intimately bound at the minor NLS-site of importin α3? The answer to this question may be unrelated to nuclear import. Prior research found Set9-mediated methylation of p65 Lys314 and Lys315 results in degradation of the activated NF-κB complex and down-regulation of NF-κB gene expression[68]. The p65 residues Lys314 and Lys315 are inaccessible in complex with importin α3, possibly preventing Set9-mediate methylation. Thus, the p65-NLS binding at the minor NLS pocket may reflect a regulatory mechanism to sequester a critical region of p65 from cellular binding partners, preventing post-translational modifications from inactivating NF-κB signaling.

**Importin α3 flexibility mediates NF-κB specificity**. Structural comparison between the p50- and p65-NLSs bound conformations of importin α1 and α3 revealed a hinge region in Arm 4 and a region of shear motion of Arm 7 of importin α3, which results in conformational changes within the major and minor NLS-pocket absent in the more rigid importin α1 (Fig. 5c, g). This observation is in line with recent work on other NLS-cargos recognized preferentially by importin α3, such as SOX2 and the Henipavirus W proteins that concluded specificity for importin α3 requires repositioning of Arm 7, which is more rigid in importin α1[24,58]. Another study found RCC1 specificity for importin α3 is due to the drastic conformational changes between Arms 1 and 4 required to accommodate a bulky β-propeller domain adjacent to the NLS[21]. Molecular dynamics simulations[7] also provided evidence that importin α3 is a flexible solenoid that undergoes segmented motion in response to cargo-binding. Precisely, the N- and C-terminal arms move as rigid bodies with respect to the central core, allowing this isoform to open and close to accommodate bulky domains flanking an NLS, as seen for RCC1[21] and SOX[58] or extend in response to elongated bipartite NLSs, as seen for Henipavirus W protein[24] and Influenza Pb2[7]. The common denominator of all importin α3 cargos is a complex structural topology, whereby the NLS is exposed onto the tertiary or quaternary structure of an oligomeric structure, as seen for the NF-κB p50/p65 heterodimer. Importin α3 conformational flexibility allows this isoform to associate with topologically complex cargos by undergoing local conformational changes that maximize NLS association. We proposed that this molecular recognition is equivalent to an induced fit for NLS-cargos.

**Mutually exclusive binding of IκBα and importin α3 to NF-κB p50/p65**. Two crystal structures of p50/p65 bound to IκBα revealed the inhibitor binds specifically to the p65-NLS[41,69], while the p50-NLS is solvent-exposed and potentially available for importin α3 binding in the inhibited state. In parallel, shown in this paper, the p50-NLS is the predominant determinant for p50/p65 binding (Fig. 3a). Our competition studies using purified IκBα, importin α3, and pre-formed complexes of the two proteins with p50/p65 (Fig. 4) revealed that not only is importin α3 unable to dissociate IκBα from p50/p65, but the import adapter also fails to associate with a pre-formed p50/p65:IκBα complex. Thus importin α3 is unlikely to shuttle the NF-κB:IκBα complex through the NPC[70], as suggested by treating unstimulated cells with the Crm1-inhibitor leptomycin B[71–73]. As previously pointed out[74], shuttling of the inhibited p50/p65:IκBα complex was only observed in the presence of leptomycin B, whereas all biochemical approaches failed to detect nuclear NF-κB:IκB complexes in unstimulated cells[74], arguing that the shuttling of the inhibited p50/p65 complex could reflect an artificial result associated with the use of leptomycin B.

Furthermore, since IκBα does not bind directly to the p50-NLS, which we found is the predominant NLS in p50/p65, the inhibitor must prevent importin α3 binding to the p50-NLS through steric hindrance, rather than direct competition for the same moiety. Overall, these data challenge the classical paradigm of NF-κB retention[75], which explains p50/p65 cytoplasmic retention as a direct consequence of masking the p65-NLS. Instead, IκBα appears to be a non-competitive inhibitor of the p50-NLS that prevents importin α3 binding to the p50/p65:IκBα complex by a structural mechanism possibly involving steric hindrance rather than direct competition for the NF-κB NLSs. Thus, NF-κB p50/p65 nuclear import is only possible after IκBα has been degraded (Fig. 9a).

**The implication of NF-κB nuclear transport for transcription**. NF-κB nuclear localization is a highly regulated import pathway, and its dysregulation is linked to several human diseases, including cancer[31]. Upon nuclear localization, the NF-κB dimers exhibit distinctly different transcriptional properties: the p50/p65 and p65/p65 complexes promote gene expression via their C-terminal transactivation domain, whereas the p50/p50, lacking a transactivation domain, suppresses gene expression. Both p65/p65 and p50/p50 homodimers are less abundant than p50/p65 in vivo and possess biological activities (activation and repression, respectively) independently regulated to ensure sustained gene expression[76]. The p65/p65 is the least stable NF-κB dimer, and while its concentration is likely low, this homodimer can also be inhibited and retained in the cytoplasm by IκBα[44] (Fig. 9b). However, less clear is why the p50/p50 homodimer is excluded from the nucleus. This dimeric transcription factor contains two potent NLSs and functions as a transcription 'repressor' due to its ability to occupy κB sites and suppress gene expression[77–79]. The data presented in this paper lead us to hypothesize that human cells evolved several different cellular mechanisms to favor p50/p65 nuclear translocation over the p50/p50 complex. First, the relative concentration of p50/p50 homodimer is low compared to the more abundant p50/p65 heterodimer due to the reduced stability of the p50/p50 dimerization interface compared to the p50/p65 heterodimer[37,38,41,42]. Second, as demonstrated in this paper, the p50-NLS binds exclusively the major NLS-binding site of importin α, which results in a large, bi-dimeric import complex p50/p50/(α/β)$_2$ (Fig. 9c), that is energetically more costly to translocate into the nucleus than smaller cargos[80,81] such as p50/p65 and p65/p65. As a result, nuclear import for cargos requiring multiple copies of importin α/β like the p50/p50 complex is likely disfavored and out-competed by cargos that require only a single importin α/β like p50/p65. Finally, NF-κB complexes maturation and NLS exposure utilize different signaling cascades, which impart energetic differences between the p50/p65 and p65/p65 complexes compared to the p105/p50 precursor complex. In the NF-κB canonical signaling pathway, the p50/p65 and p65/p65 maturation are less energetically costly than the p105/p50 maturation, which requires IKK-mediated phosphorylation, ubiquitination, and ATP-dependent proteasomal degradation of the p105 ARD domain[52]. These energetic and kinetic barriers suggest the nuclear translocation of the p50/p50 homodimer is less efficient than the p50/p65 heterodimer, ensuring appropriate gene expression and healthy cellular signaling.

In conclusion, this paper describes the mechanisms of NF-κB dimer recognition by the isoforms α3. Our work paves the way to identifying inhibitors that target the p50-NLS:importin α3 association, reducing the accumulation of NF-κB in the nucleus and dampening aberrant NF-κB signaling.

## Methods

**DNA, plasmids, and peptides**. The plasmids encoding human p65 (residues 1–306) and ΔNLS-p50 (residues 41–351) cloned in the pLM1 vector were a generous gift of Dr. Stephen Harrison[41]. ΔNLS-p65 (residues 1–290) was generated by introducing a stop codon at position D291 of p65 using the QuikChange site-directed mutagenesis kit (Stratagene). A synthetic gene encoding human p50 (residues 1–366), p50 lacking the DNA-binding domain ΔDBD-p50, residues 243–366), and p65 lacking the DNA-binding domain ΔDBD-p65, residues 191–316) were purchased from Genewiz and cloned between BamHI and XhoI restriction sites of expression vectors pGEX-6P-1 (GE Life Sciences). Plasmids encoding ΔIBB-importin α1 (residues 70–529) and ΔIBB-importin α3 (residues 68–487 and 63–487) were cloned into vectors pET-30a and pGEX-6P or pET-15b, respectively, as previously described[7,21]. Untagged IκBα (residues 67–287) was cloned in the pET-11a[65–67]. Peptides encompassing the p65-NLS (294-DRHRIEEKRKRTYETFKSIMKK-315) and p50-NLS (355-DKEEVQRKRQKLMP-368) were synthesized from GenScript (Piscataway, NJ).

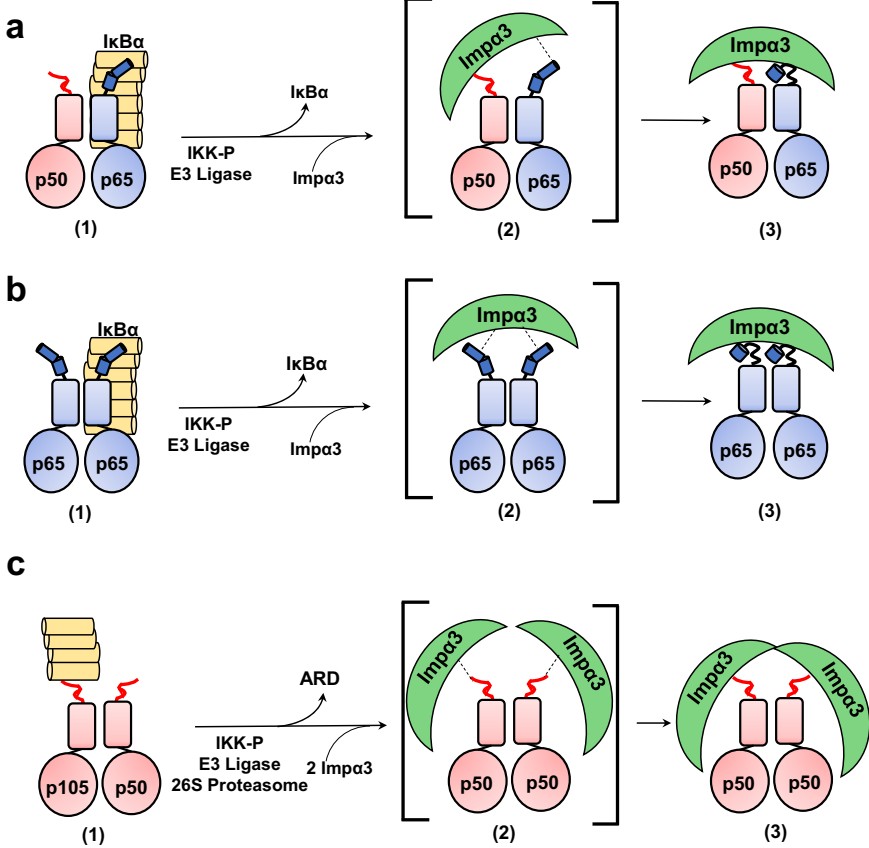

**Fig. 9 Recognition of NF-κB dimers by importin α3. a** Schematic diagram of the steps leading to exposure of NLS, binding, and nuclear import of the NF-κB complexes of **a** p50/p65, **b** p65/p65, and **c** p50/p50. On the left is the cytoplasmic conformation inhibited by IκBα through masking of the p65-NLS for p50/p65 and p65/p65 complexes or by the ARD domain masking the p50-NLS in p105/p50 complex (1). Upon activation by stimuli, the inhibited complexes are liberated by either dissociation of IκBα in p50/p65 or p65/p65 complexes or cleavage of the ARD domain in the p50/p50 complex by IKK, E3 Ligase, and 20S proteasome, which expose two NLSs for recognition by importin α3 (2). On the right is the competent import conformation recognized by importin α3 that binds the exposed NLS on the surface of the NF-κB complexes (3).

**Recombinant protein expression and purification.** Expression plasmids for all human p65, p50 (and relative mutants), ΔIBB-importin α1, ΔIBB-importin α3 were transformed and expressed in BL21 DE3 *E. coli* strain in the presence of ampicillin or kanamycin. Plasmids encoding GST-ΔDBD-NF-κB and ΔIBB-Importin α3 were co-transformed in BL21 DE3 *E. coli* strain and expressed in the presence of ampicillin and kanamycin. Bacterial cultures were grown in LB medium at 37 °C until $A_{600}$ = ~0.6 AU and all p65 induced at 16 °C with 0.25 mM IPTG for 12–16 h, all p50 induced at 18 °C at 0.25 mM IPTG for 12–16 h, and all importin α1 and α3 at 28 °C with 0.5 mM IPTG for 4 h. Cell pellets expressing all untagged p65 and p50 constructs (with or without the C-terminal NLS) were sonicated in lysis buffer 1 (20 mM HEPES, pH 7.5, 400 mM NaCl, 2 mM EDTA, 2 mM DTT, 1 mM PMSF, 5% (v/v) glycerol) followed by 0.2% polyethyleneimine precipitation for 1 h at 4 °C and 55% ammonium sulfate precipitation for 2 h at 4 °C. The p65, ΔNLS-p65, and ΔNLS-p50 ammonium sulfate pellets were re-suspended in Re-Suspension Buffer (20 mM HEPES, pH 7.5, 2 mM DTT, 2 mM EDTA, 0.2 mM PMSF, 5% (v/v) glycerol), loaded onto 5 mL HiTrap™ SP HP Column (GE Healthcare), washed with 5 CV of Buffer A (20 mM Tris, pH 7.5, 50 mM NaCl, 5 mM β-Mercaptoethanol, 2 mM EDTA, 0.2 mM PMSF), and the proteins were eluted using a linear gradient from 0% to 100% Buffer B (20 mM Tris, pH 7.5, 500 mM NaCl, 5 mM β-Mercaptoethanol, 2 mM EDTA, 0.2 mM PMSF). Cell pellets expressing GST-p50, GST-ΔDBD-p50, GST-ΔDBD-p65 and co-expressed GST-ΔDBD-NF-κB and ΔIBB-Importin α3 complexes were sonicated in Lysis Buffer 2 (20 mM Tris, pH 8.0, 250 mM NaCl, 1 mM PMSF, 5 mM BME, 0.1% (v/v) TWEEN 20, 1 mM EDTA), followed by 0.2% polyethyleneimine precipitation for 1 hour at 4 °C. GST-p50, GST-ΔDBD-p50, GST-ΔDBD-p65, and GST-ΔDBD-NF-κB:ΔIBB-Importin α3 complexes were purified by affinity chromatography on Glutathione Resin (GenScript) and incubated overnight with 100 U of PreScission Protease in PPase Cleavage Buffer (20 mM Tris–HCl, pH 7.0, 150 mM NaCl, 1 mM DTT, 1 mM EDTA, 0.1% (v/v) TWEEN 20). Untagged p50, ΔDBD-p50, ΔDBD-p65, and ΔDBD-NF-κB:ΔIBB-Importin α3 complexes were recovered off the beads. All GST-tagged[7] and His-tagged ΔIBB-Importin α1 and α3[21] were purified as previously described. IκBα was expressed and purified as previously described[65–67]. All purified samples were concentrated with a 10- or 30 kDa Millipore PES concentrator and injected on a Superdex 200 16/60 column (GE Healthcare) pre-

equilibrated with Gel Filtration Buffer 1 (20 mM Tris–HCl pH 7.5, 150 mM NaCl, 5 mM β-Mercaptoethanol, 2% (v/v) glycerol). Sample purity was confirmed by SDS–PAGE (13.5%) analysis. Complexes eluted from beads were further purified on a Superdex 200 16/60 column (GE Healthcare) pre-equilibrated with Gel Filtration Buffer 1. Co-purified ΔDBD-NF-κB:ΔIBB-Importin α3 complexes were further purified on a Superdex75 10/300 column (GE Healthcare).

**Complex assembly.** Heterodimeric p50/p65 complexes (e.g., p50/p65, p50/ΔNLS-p65, ΔNLS-p50/ΔNLS-p65) were assembled by incubating in a 1:1.5 p50:p65 molar ratio, respectively, together at 25 °C for 30 min, then at 37 °C for 1 h, followed by centrifugation at 12,000 × g at 4 °C for 30 min to remove precipitation[41]. Homodimeric NF-κB (p65/p65, p50/p50, ΔNLS-p65/ΔNLS-p65, ΔNLS-p50/ΔNLS-p50, ΔDBD-p65/ΔDBD-p65, ΔDBD-p50/ΔDBD-p50) complexes were concentrated with a 30 kDa Millipore PES concentrator and injected on a Superdex 200 16/60 column (GE Healthcare) pre-equilibrated with GF Buffer 1. All NF-κB trimeric complexes were assembled by incubating pre-formed NF-κB hetero/homodimers in a 1.5-molar excess of ΔIBB-importin α3 or IκBα, followed by incubation at 4 °C for 1 h. Complexes were concentrated with a 30 kDa Millipore PES concentrator and injected on a Superose 12 10/30 GL column (GE Healthcare) pre-equilibrated with GF Buffer 1 followed by SDS–PAGE analysis. Trimeric complexes of NF-κB p50 or p65 dimerization domains bound to importin α3 ΔDBD-p50/ΔDBD-p50:α3 and ΔDBD-p65/ΔDBD-p65:α3, Supplementary Fig. 2) were formed by co-expression in bacteria. Supplementary Table 1 reports the expected MWs for all the complexes assembled in this study.

**ELISA-based microtiter binding assay, native gel electrophoresis.** For microtiter binding assay[1], NF-κB hetero/homodimers (200 ng in 100 µL 0.05 M carbonate bicarbonate buffer, pH 9.6) were coated on ELISA plates (Nunc MaxiSorp Flat-Bottom Plate, 44-2404-21) and incubated at 4 °C overnight. The plates were blocked with PBS containing 5% skim milk for 2 h at room temperature and were washed three times with PBS containing 0.05% Tween-20 (PBST). The His-importin α samples were diluted in PBST (from 1000 to 0 nM) and 100 µL added into the wells, and the plates were incubated at room temperature for 1 h. The

plates were washed three times with PBST and incubated with anti-6xHis-4HRP-conjugated rabbit polyclonal antibody (1:3500) (Abcam ab1187) for 1 h at 37 °C. After washing with PBST three times, the reaction was initiated by the addition of 100 μL of tetramethylbenzidine (TMB) substrate (1-Step Ultra TMB-ELISA (Ref 34028), Thermo Scientific). The colorimetric reaction was stopped by adding 100 μL of 2 M $H_2SO_4$, and absorbance was read at 450 nm wavelength using a Tecan microplate reader. All experiments were carried out in duplicates. One-site specific binding analysis using least-squares fit was performed using GraphPad Prism 9 for Windows, GraphPad Software, LLC (version 9.2.0). Native gel electrophoresis was performed on a 1.5% agarose gel, as previously described[82]. In the p50/p65:α titration assay in Fig. 2, 10 μg of ΔIBB-importin α1 and ΔIBB-importin α3 were incubated with increasing concentrations of the pre-assembled p50/p65 heterodimers. For the competition assay in Fig. 4, 10 μg of p50/p65 heterodimer was incubated for one hour at 4 °C with increasing molar equivalents of ΔIBB-importin α3 or IκBα. Similarly, pre-formed trimeric p50/p65:α3 or p50/p65:IκBα complexes were challenged with increasing concentrations of either ΔIBB-importin α3 or IκBα. The mixture was separated on a 1.5% agarose gel at 4 °C for 2 h at 50 V. After electrophoresis the gel was fixed in Gel Fixing solution (25% (v/v) iso-propanol and 10% (v/v) acetic acid) for 20 min and then equilibrated with 95% (v/v) ethanol for 2 h. Gels were then dried, stained for 10 min in 0.4% (w/v) Coomassie brilliant blue R250 in Gel Fixing solvent, and destained in Gel Fixing solvent until the background was clear.

**Crystallographic methods**. His-ΔIBB-importin α1 and α3 were concentrated to ~15–20 mg ml⁻¹ and crystallized using the vapor diffusion hanging drop method by mixing 2 μl of purified protein with 0.5 μl containing a 2.5-fold molar excess of NLS peptide. The best crystals for ΔIBB-importin α3 bound to p50-NLS were obtained using 0.2 M potassium thiocyanate, 20% PEG3350, 0.1 M BIS–TRIS pH 6.5. ΔIBB-importin α3 bound to both the p50- and p65-NLS peptides was crystallized in the presence of 0.2 M ammonium sulfate, 0.1 M BIS–TRIS pH 6.5, 25% PEG 3350. For ΔIBB-importin α1, all complexes with the p50-NLS and p65-NLS alone or p50-NLS + p65-NLS peptides were crystallized in the presence of 0.5-0.65 M sodium citrate, 0.1 M HEPES pH 6.0, 10 mM β-Mercaptoethanol. All crystals were harvested in nylon cryo-loops, cryo-protected with 27% ethylene glycol, and flash-frozen in liquid nitrogen. Complete diffraction data were collected at beamlines 9-2 and 12-1 at the Stanford Synchrotron Radiation Lightsource (SSRL) on a Dectris Pilatus 6 M detector (Table 1). The structure of ΔIBB-importin α3 bound to the p50-NLS was solved by molecular replacement (MR) using the open ΔIBB-importin α3 (PDB: 6BVV) as a search model, as implemented in PHASER[83] (version 2.8). The p50-NLS was built in an unbiased Fo–Fc electron density map using Coot[84] (version 0.9.4), and the model was subjected to several rounds of real-space refinement using *phenix.real_space_refine*[85] (version 1.18.2-3874) in combination with reciprocal-space and TLS refinement using *phenix.refine*[86] (version 1.18.2-3874). This model was then used as a search template to solve the structure of ΔIBB-importin α3 bound to both the p65- and p50-NLSs, also using PHASER[83]. The p65-NLS and p50-NLS were modeled in unbiased Fo-Fc densities readily visible at the minor and major NLS-binding pockets, respectively, using Coot[84]. The complete model of ΔIBB-importin α3/p50-NLS/p65-NLS was refined as described above. The final models of ΔIBB-importin α3 bound to just the p50-NLS or both the p65- and p50-NLSs have a $R_{work}/R_{free}$ of 19.9/22.0 and 19.6/23.7, at 2.10 and 2.85 Å resolution, respectively (Table 1). For ΔIBB-importin α1, both complexes with the p50-NLS and p50/p65-NLSs were solved by molecular replacement (MR) using the rigid structure of ΔIBB-importin α1 (PDB: 3Q5U) as a search model, as implemented in PHASER[83]. Atomic models for the NF-κB NLSs were built using Coot[84] and refined using *phenix.refine*[86]. The final models of ΔIBB-importin α1 bound to p50-NLS, p65-NLSs, or both p65/p50 peptides were refined to an $R_{work}/R_{free}$ of 20.0/23.5, 19.1/21.8, and 19.8/24.1, at 2.82, 2.24, and 2.40 Å resolution, respectively. Final stereochemistry was validated using MolProbity[87]. Data collection and refinement statistics are summarized in Table 1.

**Size exclusion chromatography coupled to small angle X-ray scattering**. SEC–SAXS analysis was performed at ID7A1 station at MacCHESS, which is equipped with an AKTA Pure FPLC system (GE Healthcare). The trimeric p50/p65:ΔIBB-importin α3, p65/p65:ΔIBB-importin α3 and p50/p50:ΔIBB-importin α3 complexes were pre-assembled and concentrated at 4.5, 6.6, and 6.29 mg ml⁻¹, respectively. At the beamline, pre-formed complexes were injected on a Superdex 200 10/300 GL column (GE Healthcare) equilibrated in SEC–SAXS Buffer (20 mM Tris–HCl pH 7.0, 175 mM NaCl, 2% glycerol, and 0.5 mM TCEP). SAXS data were recorded on an EIGER 4 M detector (Dectris Ltd. Baden, Switzerland) in vacuo at 2 s per frame with a fixed camera length of 1.709 m and 10.03 keV (1.237 Å) energy allowing the collection of the angular range $q$ between 0.008 and 0.54 Å⁻¹. Primary reduction of the SAXS data was performed using RAW[88] (versions 1.6.4 and 2.0.3), and ATSAS software[89] (version 2.6.0). To minimize the effects of damaged material accumulating on the X-ray sample window and to help compensate for any baseline drift, the buffer profile was constructed by averaging the frames before the sample peak frames. The Guinier plots of the subtracted profiles were linear to the lowest measured $q$ value. GNOM[90] was used to calculate $P(r)$ plots from the scattering data. Ab initio model calculations to generate an average electron density

from solution scattering data were done using DENSS[91], as implemented in RAW. The DENSS densities have a Fourier shell correlation (FSC) of 52.2 Å (p50/p65/ΔIBB-importin α3), 64.7 Å (p50/p50/ΔIBB-importin α3) and 50.7 Å (p65/p65/ΔIBB-importin α3), respectively. Docking of PDB models inside the SAXS density was done manually and improved by rigid-body refinement using Chimera[92]. Theoretical solution scattering curves were calculated using the FoXS web server[93]. SEC–SAXS data collection and analysis statistics are in Supplementary Table 3.

**Structure analysis and modeling**. All structural illustrations were generated using PyMoL (Schrödinger, Inc.) and Chimera[92]. Binding interfaces were analyzed using PISA[55] and PDBsum[94]. Secondary structure superimpositions were done in Coot[82] and analyzed using SuperPose 1.0 server[59]. DynDom[56] was used to identify importin α3 domain movements upon cargo-binding.

**Reporting summary**. Further information on research design is available in the Nature Research Reporting Summary linked to this article.

## Data availability
Coordinates and structure factors for the ΔIBB-importin α3 + p50-NLS, ΔIBB-importin α3 + p50-NLS + p65-NLS and ΔIBB-importin α1 + p65-NLS, ΔIBB-importin α1 + p50-NLS, and ΔIBB-importin α1 + p50-NLS + p65-NLS complexes have been deposited in the Protein Data Bank (accession codes 7LFC, 7LF4, and 7LEU, 7LEQ, 7LET). Additionally, all other PDBs analyzed in this study include 1NFK, 1RAM, 1VKX, 6BVV, 3Q5U, 6BVZ, 6WX8, and 5TBK. SEC–SAXS and all other data are available from the corresponding author upon reasonable request. Source data are provided with this paper.

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

## Acknowledgements

We are thankful to the staff at SSRL beamlines 9-2 and 12-1 and MacCHESS for beamtime at the ID7A1 station and data collection assistance. We also thank Dr. Elizabeth Komives at UCSD for the generous gift of the IκBα expression plasmid. This work was supported by the National Institutes of Health grants R01 GM122844, R35 GM140733, S10 OD017987, and S10 OD023479 to G.C. Research in this publication includes work carried out at the Kimmel Cancer Center X-ray Crystallography and Molecular Interaction Facility at Thomas Jefferson University, which is supported in part by the National Cancer Institute Cancer Center Support Grant P30 CA56036. CHESS is supported by the National Science Foundation grant DMR-1829070, and the MacCHESS resource is supported by the National Institutes of Health grant P30 GM124166-01A1 and NYSTAR.

## Author contributions

T.J.F., R.K.L., D.P.Y., R.S.S., and G.C. conceived the project, planned the experiments, and analyzed the data with help from C.O. All SEC–SAXS experiments were performed by R.E.G. who also helped analyzing the data. T.J.F. and G.C. wrote the paper with help from all other authors.

## Competing interests

The authors declare no competing interests.
