## [Peer Review File · Nature Communications]

REVIEWER COMMENTS

Reviewer #1 (Remarks to the Author):

The manuscript submitted by Florio et al., describes the structural basis for NFkB binding to the nuclear import adapter, importin alpha. The major claims of the paper include a mechanistic elucidation of NFkB binding to specific isoform adapters (such as importin alpha 3), using a range of structural and biophysical approaches. The manuscript describes how the p65:p50 heterodimer binds preferentially to importin alpha 3 by forming a bipartite NLS in trans. Specifically, they show that p50 binds at the major site, while p65 binds at the minor site. The mechanistic basis for this specificity represents a highly novel breakthrough. This mechanism has not been described before in nucleocytoplasmic transport, and provides an understanding for how NFkB, an important cargo that regulates innate immunity, is directed to the nucleus following appropriate signalling events. How heterodimers can bind across the major and minor sites of the importin alpha adapter through mimicking a bipartite NLS, is highly novel. I believe that this research will be of very high interest to researchers in cell trafficking, structural biology, and immunology. The work is convincing, and based on many high-resolution, and the results are confirmed using a range of biophysical assays and SAXS. The level of detail provided in the manuscript would allow the research to be reproduced. While the general presentation of the manuscript is good, there are a number of issues that need to be addressed.

1. Line 18-19. Importin-beta is referred to as the receptor. For consistency, importin-alpha should be referred to as the adapter.
2. Line 23. Whenever referring to armadillo, this should be followed by motif. This will assist the readability, particularly for those not expert in structural biology or solenoid proteins containing this motif.
3. It is quite unusual for a helical motif to be present in NLSs. Generally these are in extended conformations. The authors have commented on this twice in the manuscript, however a description of how unusual this is should be made clearer.
4. Line 26. "live cells" should be replaced with in vivo or equivalent.
5. Line 36. "At the essence of nuclear transport...." Consider reworking this sentence.
6. Line 68. "Besides" is not needed here.
7. Figure 1A. It would be highly informative to also include the NF-kB subunit domain structures and NLS sequences for RelB, c-Rel, and p52. This does need to (or rather shouldn't) be in the main figure, but should be included as a supplementary figure. Having all 5 NF-kB molecules aligned would allow readers to more easily appreciate how the results may inform IMPA binding in these other immune signalling molecules.

8. Line 94 needs a rework: Structural studies revealed that the p65:p50 dimerization interface is stabilized by a network of stabilizing contacts,

9. Line 132. Don't need #

10. Line 147. E.coli should be E. coli

11. Line 294. "We then assembled and purified three NF- κ B dimers (Fig. 1B)". However the figure refers to only a cartoon.

12. Figure 1. Is there an explanation for why IMPA1 and IMP3 run differently on native page? Were the gels run for the same length of time?

13. Figure 4. It would be more informative if H-bond and salt bridges could be labelled. That way the reader can confirm / investigate the 18 H-bonds etc that have been described in the manuscript, and compare how these differ between IMPA1 and 3. Some H-bonds have been added as dotted lines, and although it is mostly intuitive where the other H-bonds may be, it would be more informative to have all H-bonds and salt bridges clearly demarcated,

14. Figure 4G. I believe this is likely to be mis-labelled. Currently, it's labelled as IMPA3, however the residue numbering would suggest it's IMPA1. It is important that this is checked as it would be very confusing for readers that are not familiar with the numbering differences between IMPA1 and IMPA3.

15. Supplementary Figure 1. Can more information be added to the figure legend. Currently, it indicates that the proteins were SEC assembled, however the manuscript text indicates the proteins were co-expressed. If they are co-expressed, it infers that the proteins were co-purified during affinity chromatography. The procedure needs to be clarified in the figure legend and/or the manuscript text.

16. Supplementary Figure 3. Can the NLS sequences be added to the figure, and these residues added to the figures? This would greatly assist the readability for non-structural biologists. Also, is there any reason for the different shades of green? I believe panel C is mis-labelled. Should 7LET be the heterodimer? Otherwise, what's the difference between panels A and C? According to the Table of crystallographic data, C should be the heterodimer.

Reviewer #2 (Remarks to the Author):

Authors T.J. Florio, *et al*. report a structural, biophysical, and biochemical investigation into the molecular nature of the interaction of nuclear localization sequence (NLS) polypeptide regions of NF- κ B transcription factors and importin- α receptor subunits of active nuclear import complexes. The study addresses longstanding mechanistic questions about nucleocytoplasmic shuttling of NF-

κ B. In absence of binding to cytoplasm I κ B proteins, NF- κ B rapidly migrates to the nucleus, through preferential association with the importin- α 3. As NF- κ B is a dimer of structurally related protein subunits, both of which contain similarly positioned NLS polypeptides of diverse amino acid sequence, it has remained in question whether either of the two NLSs is necessary, sufficient, or if they are functionally redundant with regards to nuclear interaction. There also remain questions as to whether the distinct NF- κ B NLS polypeptides interact differently with closely related importin- α proteins.

The authors approach these questions with a straightforward study analyzing the binding interaction with purified, recombinant proteins *in vitro*. They support their solution binding studies with structural characterization, via x-ray crystallography, of p50 and p65 NLS peptide:importin- α 1/ α 3 complexes and size-exclusion chromatography/small-angle x-ray structure analyses of NF- κ B:importin- α 3 complexes. The results show a clear preference of the distinct p50 and p65 NLS peptides for the two binding sites in importin- α 3 and propose a mechanism where, by binding the two NLS polypeptides as if they are one bipartite NLS, the importin- α 3 receptor preferentially targets NF- κ B p50:p65 heterodimer for nuclear localization. The experimental work is well executed and extremely well presented. The conclusions are sound and, with the exception of a few questions and suggested minor edits listed below, the manuscript is well written. While not earth shattering, the report fills a void in the literature on NF- κ B regulatory mechanisms.

Questions:

1) In lines 90-98 of the Introduction, the authors do an admirable job of summarizing the well established work on stability of NF- κ B dimer interfaces. However, after describing how the Asp254:Asn200 hydrogen bond provides maximum stability to the p50:p65 heterodimer and that repulsion of Asp254 residues contributes to the relative instability of p50:p50, the authors leave the reader to assume incorrectly that p65:p65 would then be more stable than p50:p50. This is not true and it needs to be mentioned that, due to other factors (fewer H bonds and less intimate dimer interaction in which the Asn200 residues do not come into contact with one another) the p65:p65 homodimer is even less thermodynamically stable in solution than p50:p50.

2) I am very intrigued by the observed binding dissociation constant for the p65:p65 homodimer:importin- α 3 complex, reported by ELISA at \sim 57 nM. As stated (lines 312-331), this is significantly higher binding affinity than p50:p65 heterodimer (\sim 324 nM). Furthermore, the Δ NLS-p50:p65 heterodimer fails to bind. In the Discussion, there is a good deal of space dedicated to the role of the p65 NLS in importin- α binding and other roles (I κ B, Set9, etc.) for this region. But I don't see a clear suggestion as to why the affinity of p65:p65 for importin- α 3 should increase so dramatically as a consequence of a second p65 NLS. Could that really be the result of increased avidity alone? Or, perhaps, the close proximity of two p65 NLSs affects their respective structures and/or dynamics asymmetrically in ways that promote/enhance their binding? One can envision

complicated experiments involving single point mutations of p50 NLS with and without p65 NLS on the p65 subunit or even generation of p65NLS:mutant p65NLS "heterodimers". While such studies are beyond the scope of the present study, which is complete as is, it would be interesting to hear more from the authors on this point specifically in the Discussion.

3) One other set of potentially relevant experimental observations that is not referenced in the Discussion involves background nucleocytoplasmic shuttling of inactive p50:p65:I κ B α complexes through the p50 subunit NLS irrespective of inducing signal. This was shown back in the late 1990s by a number of labs (Shigeki Miyamoto, Tom Hope, others) by treatment of cells with leptomycin B and subsequent observation of "trapped" nuclear NF- κ B:I κ B α complexes. Is this a valid experiment? It seems to fit within your model that p50 is sufficient to promote nuclear localization of NF- κ B dimers.

4) There are some issues with the citations of references throughout. On line 100 of the Introduction section, for example, both references "22,23" and then "21,22" are cited at different parts of the same sentence. This is odd. Furthermore, it is not completely clear why reference 23 is being invoked here. Also, on line 405 there is citation of references "43,444". While this is clearly a typo, it begs the question as to whether these are indeed the correct references for the studies that are cited in the sentence (I don't see reference to the I κ B β :p65 homodimer crystal structure report, for example, though it is mentioned in that sentence). In general, the authors should review their references throughout and make certain that they are properly cited in the manuscript.

Suggested minor edits:

Introduction

Line 74: The word "state" should be "states"

Line 79: It is a bit confusing to designate the NF- κ B p50 subunit with the parenthetical "(p105 precursor)". It should be made clearer that p50 is the partially digested product of the p105 precursor protein, or something to that effect. Likewise for the relationship of p52 to p100.

Lines 88-90: Your Greek " κ " switches to "k" a couple of times in this sentence.

Line 95: It feels like a verb is missing in the portion of the sentence that begins with, "...among which..."

Materials and Methods

Line 147: I suspect that it should read, "BL21 DE3"

Results

Lines 312-313: I don't believe that "co-expressed" correctly describes the experimental protocol here. Perhaps, "combined"?

Lines 390-391: "van der Wal" should be "Van der Waals".

Discussion

Line 543: Should "determinants" be "determinant"?

Line 633: "NF-kB" should be "NF-κB"

Figure Legend

Figure 1: "highlighted" is in the wrong place and there is a "k" in "IkB α ".

Figures

Figure 3, panel B: For consistency with Table 1 and throughout the rest of the manuscript, this should be titled "p50/p65-NLS: α 3" not "p50:p65-NLS: α 3".

Reviewer #3 (Remarks to the Author):

This manuscript uses extensive structural techniques and some biochemical techniques to reveal how the p50 and p65 NLSs bind the importin $\alpha 3$ major and minor pockets, respectively, generating a trans bipartite NLS. The authors propose that p65 NLS binding at the minor pocket induces a conformational change in the importin $\alpha 3$ Arm-core and that this plays a role in NLS cargo specificity, particularly importin $\alpha 3$'s preference for p65:p50 and ability to discriminate between p50:p50 and p65:p65 homodimers. On the other hand, importin $\alpha 1$ p50, p65 and p65/50 bound complexes superimpose with low RMSD and are reported as rigid.

Given mechanisms underlying NF- κ B nuclear import are important in regulating the expression of genes involved in numerous processes, this study is potentially of broad-interest. However, overall, I have some concerns about the robustness of some data that lays the foundation of some of the key observations in the paper. Also, in some cases there is also not enough detail provided to adequately interpret the data.

Major points:

1. The authors use an ELISA-based binding assay to compare the binding affinities, of Δ IBB-importin $\alpha 3$ for NF- κ B dimers containing one or two NLSs. These measurements form a key part of the paper. There are a few issues. 1) The authors compare binding measurements from Figure 2E with measurements obtained from Supplemental Figure 2. If one is comparing values between samples, they should be run in the same assay. 2) The green and red curves in Figure 2E have not reached equilibrium. 3) To derive reliable binding measurements, both time and concentration should be varied to establish equilibration. Based on the above points the K_D values reported are not reliable, and are merely an indication of whether binding has or has not taken place. Preferably, a technique such as SPR, and experiments that consider the above points, should be used.

Also, in Supplementary Figure 2B there are more data points for Δ NLS-p50 and Δ NLS-65 than for p50 and p65. Have data points been removed from the curve?

2. 7LF4 has two molecules in the asymmetric unit – is the importin $\alpha 3$ dimer functionally significant? Was the p50 and p65 NLS bound to both importin $\alpha 3$ molecules? If so, how did the binding/conformation of the p50 and p65 NLS compare between the two molecules? Did nearby

crystal contacts influence the conformation of any bound peptides? When comparing the two importin $\alpha 3$ molecules in the asymmetric, what is the RMSD? Please note the molecule that was analysed in the study.

3. Line 426 – ‘importin $\alpha 3$ adopts a significantly different conformation when the p65-NLS is bound at the minor NLS-box’. The difference portrayed in Figure 3C does not appear ‘significant’. What is the shift in Angstrom? Could the shift be due to differences in the resolution of the structures compared? DynDom analysis was then used to predict that importin $\alpha 3$ undergoes two conformational changes upon p65-NLS recognition at the minor site -the first, a hinge movement within Arm 4, and the second a shear movement within Arm 7. Besides Supplementary Figure 4, no further description of this movement is provided. The authors conclude that the segmented motion of the Arm core (described in line 429 as drastic) allows the minor site to become more accessible, accommodating the partially helical p65-NLS – however the authors provide no further evidence of this statement, a fundamental one on which the paper is based.

4. The movement described above also appears to have been reported elsewhere for different NLS-cargo that are preferentially recognised by importin $\alpha 3$ (as reported by the authors in the discussion paragraph beginning at line 580).

5. Please clarify line 237, with regards to the ‘docking’ of the p50-NLS peptide. ‘The p65-NLS was readily modeled in an unbiased Fo-Fc density, while the density for the p50-NLS could be DOCKED at the major NLS-pocket using Coot’.

5. When RMSD values are reported, please note what residues were aligned in the analysis. For example, in line 360 it is reported that in the two complexes the p50-NLS were superimposable (RMSD 1.095). Were only residues that could be confidently modelled into the electron density compared (see point 6 below)?

Minor points

1. Figure 2E, Supplemental Figure 2 – how many replicates?

2. Please define the residues that constitute Δ IBB-importin $\alpha 3$ and Δ IBB-importin $\alpha 1$.

3. In Figure 1A please number the regions spanning each domain.

4. Figure 2D lacks the sizes of the MW marker and therefore one can't correlate the sizes of Δ IBB-a3 and Δ DBD-p50 between this gel and that in Supplementary Fig 1A and B. On this note, I would consider it essential to note the expected MWs of all of the complexes in a table in order to fully evaluate the data.

5. Δ NLSp50 is reported as residues 41-351. Why are the first 40 residues also missing?

6. Line 350 it is reported that 'All crystal structures had clear electron densities for the p50- and p65-NLSs that we modeled in unbiased F0-Fc electron density maps and confirmed in Polder maps'. In the true definition of the term, both NLS's are entirely visible, but the NLS is defined beyond these key residues in this study (ie p50-NLS is defined as 355-368), and these 'flanking residues' are not visible in the electron density portrayed in Fig 3D and supp. Fig 3A.

Point-by-point Response to Reviewers' comments

We thank the reviewers and the editor for the valuable comments. We have extensively revised the paper to respond to reviewers' comments. All revisions in the text and supplementary info are marked in red.

Reviewer #1

There are a number of issues that need to be addressed.

1. Line 18-19. Importin-beta is referred to as the receptor. For consistency, importin-alpha should be referred to as the adapter.

Done: see line 18.

2. Line 23. Whenever referring to armadillo, this should be followed by motif. This will assist the readability, particularly for those not expert in structural biology or solenoid proteins containing this motif.

Updated: see lines 50-51. Throughout the paper, the abbreviation 'Arm' refers to 'Armadillo motif'

3. It is quite unusual for a helical motif to be present in NLSs. Generally these are in extended conformations. The authors have commented on this twice in the manuscript, however a description of how unusual this is should be made clearer.

Updated: we revised the Discussion to include a description of the helical NLSs: see lines 607-612.

4. Line 26. "live cells" should be replaced with in vivo or equivalent.

We removed 'live cells' to shorten the abstract to 150 words.

5. Line 36. "At the essence of nuclear transport...." Consider reworking this sentence

Updated: see lines 34-35.

6. Line 68. "Besides" is not needed here.

Removed.

7. Figure 1A. It would be highly informative to also include the NF-κB subunit domain structures and NLS sequences for RelB, c-Rel, and p52. This does need to (or rather shouldn't) be in the main figure, but should be included as a supplementary figure. Having all 5 NF-κB molecules aligned would allow readers to more easily appreciate how the results may inform IMPA binding in these other immune signalling molecules.

Done: see the revised **Supplemental Figure 1A**.

8. Line 94 needs a rework: Structural studies revealed that the p65:p50 dimerization interface is stabilized by a network of stabilizing contacts,

Rephrased and expanded in response to reviewer #2: see lines 93-100.

9. Line 132. Don't need #

Updated.

10. Line 147. E.coli should be E. coli

Updated throughout the text.

11. Line 294. "We then assembled and purified three NF-κB dimers (Fig. 1B)". However the figure refers to only a cartoon.

Updated: see line 312. We refer to all purified NF-κB complexes shown in **Supplementary Figure 1B**.

12. Figure 1. Is there an explanation for why IMPA1 and IMP3 run differently on native page? Were the gels run for the same length of time?

Good question. The gels were run at the same time and voltage, but it is possible that a slight difference in isoelectric point, 4.50 for Δ I β B- α 3 versus 4.97 for Δ I β B- α 2, results in slightly faster migration of the more acidic isoform α 3.

13. Figure 4. It would be more informative if H-bond and salt bridges could be labelled. That way the reader can confirm / investigate the 18 H-bonds etc that have been described in the manuscript, and compare how these differ between IMPA1 and 3. Some H-bonds have been added as dotted lines, and although it is mostly intuitive where the other H-bonds may be, it would be more informative to have all H-bonds and salt bridges clearly demarcated.

Done. In **Figure 6**, we have now labeled residues and side chains involved in hydrogen bonding and salt bridges and updated the figure legend (lines 780-786).

14. Figure 4G. I believe this is likely to be mis-labelled. Currently, it's labelled as IMPA3, however the residue numbering would suggest it's IMPA1. It is important that this is checked as it would be very confusing for readers that are not familiar with the numbering differences between IMPA1 and IMPA3.

Thank you!! The reviewer is correct, and **Figure 6G** has been relabeled. Furthermore, for the sake of clarity, we are now referring to importin α complexes with NLS-peptides by using a '+'. For instance, a complex of importin α 3 bound to p50-NLS and p65-NLS is labeled as "p50-NLS + p65-NLS + α 3". This label is less ambiguous than using a slash, which could be mistaken for the full-length p50/p65 heterodimer. Also, the '+' sign means that NLS-peptides were added to importin α without additional purification. See revised **Figs. 5,6** and **Table 1**.

15. Supplementary Figure 1. Can more information be added to the figure legend. Currently, it indicates that the proteins were SEC assembled, however the manuscript text indicates the proteins were co-expressed. If they are co-expressed, it infers that the proteins were co-purified during affinity chromatography. The procedure needs to be clarified in the figure legend and/or the manuscript text.

Done. We now clearly describe complexes that were co-expressed (**Supplementary Fig. 2** legend) versus assembled from individual components (**Supplementary Fig. 1** legend). We also included this information in the Methods (see section '**Complex assembly**', on line 186).

16. Supplementary Figure 3. Can the NLS sequences be added to the figure, and these residues added to the figures? This would greatly assist the readability for non-structural biologists. Also, is there any reason for the different shades of green? I believe panel C is mis-labelled. Should 7LET be the heterodimer? Otherwise, what's the difference between panels A and C? According to the Table of crystallographic data, C should be the heterodimer.

Done. The NLS sequences for each polder map have been added. Furthermore, we reorganized the previous Supplemental Figure 4 into two **Supplementary Fig. 3** (for importin α 3) and **Supplementary Fig. 4** (for importin α 1), using the same color-coding, as in **Figure 5**.

Reviewer #2

1) In lines 90-98 of the Introduction, the authors do an admirable job of summarizing the well established work on stability of NF- κ B dimer interfaces. However, after describing how the Asp254:Asn200 hydrogen bond provides maximum stability to the p50:p65 heterodimer and that repulsion of Asp254 residues contributes to the relative instability of p50:p50, the authors leave the reader to assume incorrectly that p65:p65 would then be more stable than p50:p50. This is not true and it needs to be mentioned that, due to other factors (fewer H bonds and less intimate dimer interaction in which the Asn200 residues do not come into contact with one another) the p65:p65 homodimer is even less thermodynamically stable in solution than p50:p50.

Absolutely, we revised the text to explain this concept, see lines 93-100. We clearly state: "Thus, NF- κ B p50/p65 is more stable than p50/p50, and the p65/p65 homodimer is the least stable of the three NF- κ B dimers".

2) I am very intrigued by the observed binding dissociation constant for the p65:p65 homodimer:importin- α 3 complex, reported by ELISA at ~57 nM. As stated (lines 312-331), this is significantly higher binding affinity than p50:p65 heterodimer (~324 nM). Furthermore, the Δ NLS-p50:p65

heterodimer fails to bind. In the Discussion, there is a good deal of space dedicated to the role of the p65 NLS in importin- α binding and other roles ($I\kappa B$, Set9, etc.) for this region. But I don't see a clear suggestion as to why the affinity of p65:p65 for importin- $\alpha 3$ should increase so dramatically as a consequence of a second p65 NLS. Could that really be the result of increased avidity alone? Or, perhaps, the close proximity of two p65 NLSs affects their respective structures and/or dynamics asymmetrically in ways that promote/enhance their binding? One can envision complicated experiments involving single point mutations of p50 NLS with and without p65 NLS on the p65 subunit or even generation of p65NLS:mutant p65NLS "heterodimers". While such studies are beyond the scope of the present study, which is complete as is, it would be interesting to hear more from the authors on this point specifically in the Discussion.

We thank the reviewer for bringing this up! In response to the reviewer #3's criticism, we have dramatically improved our binding assays, which are now presented in a standalone **Figure 3** and described in a Results section entitled "**The p50-NLS in p50/p65 is the dominant binding determinant for importin $\alpha 3$** " (on page 8). We now have a much more accurate determination of the equilibrium binding affinities of Δ IBB-importin $\alpha 3$ for p65/p50 and p65:p65, which are $K_D = 45.8 \pm 0.9$ nM (**Fig. 3A**) and $K_D = 35.2 \pm 0.9$ nM (**Fig. 3B**), respectively, with approximately the same Bmax. While it is clear from our data that the p65/p65 contains an active NLS, its limited stability and the fact $I\kappa B\alpha$ can retain the p65/p65 homodimer in the cytoplasm (same as the p50/p65) are perhaps the reasons why the cell does not effectively import this species. The Discussion was expanded to clarify this point (see lines 682-684).

3) One other set of potentially relevant experimental observations that is not referenced in the Discussion involves background nucleocytoplasmic shuttling of inactive p50:p65: $I\kappa B\alpha$ complexes through the p50 subunit NLS irrespective of inducing signal. This was shown back in the late 1990s by a number of labs (Shigeki Miyamoto, Tom Hope, others) by treatment of cells with leptomycin B and subsequent observation of "trapped" nuclear NF- κB : $I\kappa B\alpha$ complexes. Is this a valid experiment? It seems to fit within your model that p50 is sufficient to promote nuclear localization of NF- κB dimers.

Done. The review points out a significant gap in our paper that we decided to fill in the last three months. We have carried out a new set of competition experiments shown in a new **Figure 4**, and described in the Result section 'Competition between $I\kappa B\alpha$ and importin $\alpha 3$ ' (line 361) and discussed in 'Mutually exclusive binding of $I\kappa B\alpha$ and importin $\alpha 3$ to p50/p65' (line 649). We essentially observe that importin $\alpha 3$ cannot bind the $I\kappa B\alpha$ -inhibited complex, ruling out a model whereby the inhibited NF- κB is shuttled back and forth into the nucleus.

4) There are some issues with the citations of references throughout. On line 100 of the Introduction section, for example, both references "22,23" and then "21,22" are cited at different parts of the same sentence. This is odd. Furthermore, it is not completely clear why reference 23 is being invoked here. Also, on line 405 there is citation of references "43,444". While this is clearly a typo, it begs the question as to whether these are indeed the correct references for the studies that are cited in the sentence (I don't see reference to the $I\kappa B\beta$:p65 homodimer crystal structure report, for example, though it is mentioned in that sentence). In general, the authors should review their references throughout and make certain that they are properly cited in the manuscript.

Fixed. We sincerely apologize for this shortcoming. Our Endnote library was corrupted, and we did not manually check all references before submission. We've now fixed all citations and carefully reviewed each reference.

Suggested minor edits:

(Introduction) Line 74: The word "state" should be "states"

Updated. See line 72

Line 79: It is a bit confusing to designate the NF- κB p50 subunit with the parenthetical "(p105 precursor)". It should be made clearer that p50 is the partially digested product of the p105 precursor protein, or something to that effect. Likewise for the relationship of p52 to p100.

Done. The text has been revised on lines 76-78.

Lines 88-90: Your Greek " κ " switches to "k" a couple of times in this sentence.

Done. The entire text has been revised to ensure NF- κB or $I\kappa B\alpha$ has a Greek ' κ ' instead of a regular 'k'.

Line 95: It feels like a verb is missing in the portion of the sentence that begins with, "...among which..."
Updated. We revised lines 93-100.

(Materials and Methods) Line 147: I suspect that it should read, "BL21 DE3"
Updated.

(Results) Lines 312-313: I don't believe that "co-expressed" correctly describes the experimental protocol here. Perhaps, "combined"?

The text is actually correct, but we failed to explain this point in the text. In addition to assembling the NF- κ B:importin complexes from individual subunits (**Supplementary Fig. 1B**), we also successfully co-expressed NF- κ B p50 and p65 dimerization cores with importin α isoforms (**Supplementary Fig. 2**). We also updated the Material and Methods section 'Complex assembly' (line 186).

Lines 390-391: "van der Wal" should be "Van der Waals".

Fixed. We apologize for the inaccuracy. We consistently fixed all incorrect 'van der Waals' to 'Van der Waals'.

(Discussion) Line 543: Should "determinants" be "determinant"?

Fixed.

Line 633: "NF-kB" should be "NF- κ B"

Fixed: All NF- κ B and I κ B α have a Greek ' κ ' instead of a regular 'k' throughout the text.

Figure Legend

Figure 1: "highlighted" is in the wrong place and there is a "k" in "IkBa".

Fixed.

Figure 3, panel B: For consistency with Table 1 and throughout the rest of the manuscript, this should be titled "p50/p65-NLS: α 3" not "p50:p65-NLS: α 3".

Revised. We have standardized the NF- κ B subunit nomenclature to best match the literature: all NF- κ B dimers are denoted with a 'slash' between subunits and p50 precedes p65, namely, p50/p65, p50/p50, p65/p65. All complexes of NF- κ B with importin α or I κ B α are indicated by a colon, namely p50/p65: α 3, p50/p65:I κ B α , etc. Peptides added to importin α are indicated by a '+'.

Reviewer #3

Reviewer #3

1. The authors use an ELISA-based binding assay to compare the binding affinities, of Δ IBB-importin α 3 for NF- κ B dimers containing one or two NLSs. These measurements form a key part of the paper. There are a few issues.

The reviewer is 100% correct. The ELISA measurements presented in the first version of this paper were just poor: the weakest part of our paper. We apologize for the shortcoming. We have repeated all binding experiments more than a dozen times, using different ranges of importin α 3 concentration, different quantities of NF- κ B immobilized per well, different antibody dilutions, and varying the amount of detergent in the blocking solution. The new data are in a standalone **Figure 3** instead of being fragmented between Main and Supplementary figures, as in the previous version.

1) The authors compare binding measurements from Figure 2E with measurements obtained from Supplemental Figure 2. If one is comparing values between samples, they should be run in the same assay.

Given the number of constructs screened in this papers (e.g., eight different NF- κ B dimers, all purified to homogeneity, see **Supplementary Fig. 1B**), we were forced to carry out two independent sets of binding experiments (each in triplicates): in **Fig. 3A** we report the microtiter binding assay for p50/p65, p50/ Δ NLS-p65, Δ NLS-p50/p65, and Δ NLS-p50/ Δ NLS-p65; in **Fig. 3B** we report the microtiter binding assay for p50/p50, p65/p65, Δ NLS-p50/ Δ NLS-p50, and Δ NLS-p65/ Δ NLS-p65. The two sets of experiments are done under similar conditions and, though non-identical - as the reviewer pointed out - can be qualitatively compared. This critical point is now

clearly emphasized in a new Results section “*The p50-NLS in p50/p65 is the dominant binding determinant for importin $\alpha 3$* ”, on page 8.

2) The green and red curves in Figure 2E have not reached equilibrium.

3) To derive reliable binding measurements, both time and concentration should be varied to establish equilibration. Based on the above points the K_D values reported are not reliable, and are merely an indication of whether binding has or has not taken place. Also, in Supplementary Figure 2B there are more data points for Δ NLS-p50 and Δ NLS-65 than for p50 and p65. Have data points been removed from the curve?

All new binding isotherms in **Fig. 3** used to calculate K_D s are fully equilibrated. The concentration of Δ IBB-importin $\alpha 3$ was varied between 0-1000 nM, and we also tested different antibody dilutions. All K_D s determined in these new experiments are accurate, as demonstrated by the low standard errors (over three replicates). In the first set of experiments (**Fig. 3A**), we found that Δ IBB-importin $\alpha 3$ bound p50/p65 and p50/ Δ NLS-p65 with $K_D = 45.8 \pm 0.9$ nM and $K_D = 59.6 \pm 2.2$ nM, respectively, while NF- κ B dimers lacking the p50-NLS (e.g., Δ NLS-p50/p65 and Δ NLS-p50/ Δ NLS-p65) had non-saturable binding. In the second set of experiments (**Fig. 3B**), we found that Δ IBB-importin $\alpha 3$ bound p50/p50 and p65/p65 with $K_D = 15.4 \pm 0.2$ nM and $K_D = 35.2 \pm 0.9$ nM, respectively. These K_D s confirm the p50-NLS is dominant, but we also observed the p50/ Δ NLS-p65 had a slightly reduced B_{max} compared to the p50/p65 (**Fig. 3A**), possibly suggesting a reduced number of binding sites at equilibrium. We also observed that the p50/p50 has double B_{max} compared to the p65/p65, possibly suggesting two copies of Δ IBB-importin $\alpha 3$ binds the p50/p50 (the stoichiometry of p50/p50: $\alpha 3$ is explored in the SEC-SAXS section). **Preferably, a technique such as SPR, and experiments that consider the above points, should be used.** Unfortunately, our Biacore 3000 is down, and we could not repair it on time to perform binding assays for this paper.

2. 7LF4 has two molecules in the asymmetric unit – is the importin $\alpha 3$ dimer functionally significant?

Yes, there are two importin $\alpha 3$ in the crystal structure (Chain A and C) (7LF4 model).

We believe the dimer is a crystallographic artifact, not physiologically relevant. Importin $\alpha 3$ isoform does not form a dimer when bound to importin β (our unpublished cryo-EM data).

Was the p50 and p65 NLS bound to both importin $\alpha 3$ molecules?

Yes, both importin $\alpha 3$ molecules in the asymmetric unit are bound to the p50- and p65-NLS peptides.

If so, how did the binding/conformation of the p50 and p65 NLS compare between the two molecules?

The p65-NLS and p50-NLS occupy identical binding sites with similar orientations in both importin $\alpha 3$ molecules. However, one of the importin $\alpha 3$ molecules (Chain A) has a higher B-factor, likely due to weaker crystal contacts.

Did nearby crystal contacts influence the conformation of any bound peptides?

No, but the lack of crystal contacts for one importin $\alpha 3$:NLS complex (chain A) makes it less visible than the other in chain C, which we used for structural analysis.

When comparing the two importin $\alpha 3$ molecules in the asymmetric, what is the RMSD?

The two importin $\alpha 3$ molecules in Chain A and C were similar with an RMSD = 1.14 Å.

Please note the molecule that was analysed in the study.

Done, see lines 769-770.

3. Line 426 – ‘importin $\alpha 3$ adopts a significantly different conformation when the p65-NLS is bound at the minor NLS-box’. The difference portrayed in Figure 3C does not appear ‘significant’. What is the shift in Angstrom? Could the shift be due to differences in the resolution of the structures compared?

Yes, importin $\alpha 3$ Arm 10 bound to the p65-NLS shifts compared to apo importin $\alpha 3$ and p50-NLS-bound importin $\alpha 3$ by 3.5 Å and 2.3 Å, respectively (this is shown in a new **Supplemental Figure 5**). We have revised the text (lines 482-489) to clarify this point. We do not believe these differences result from the different resolutions of the three structures (within 0.5 Å) as we observe excellent electron density for the importin $\alpha 3$ backbone and its sidechains. Also, the movements are consistent with previous MD simulations (see Pumroy et al, 2015 Structure).

DynDom analysis was then used to predict that importin $\alpha 3$ undergoes two conformational changes upon p65-NLS recognition at the minor site -the first, a hinge movement within Arm 4, and the second a shear movement within Arm 7. Besides Supplementary Figure 4, no further description of this movement is provided. The authors conclude that the segmented motion of the Arm core (described in line 429 as drastic) allows the minor site to become more accessible, accommodating the partially helical p65-NLS

– however the authors provide no further evidence of this statement, a fundamental one on which the paper is based.

This is correct; Arm 4 and Arm 7 both undergo conformational changes; however, the reviewer may be mistaking the relationship between the hinge movement observed at the major NLS-binding pocket and the p65 NLS. The p65-NLS does not influence the hinge movement in Arm 4; instead, the p50-NLS is responsible for the observed conformational change in both the p50-NLS:importin α 3 and p65/p50-NLS:importin α 3 complexes compared to the apo-Importin α 3 (line 487). We have added a new **Supplementary Figure 5** describing the range of conformational motion identified by DynDom, further illustrating the conformational changes in importin α 3 induced by NLS-binding. We also expanded the Results section "**A conformational change in importin α 3 Arm-core upon NF- κ B NLS recognition**" (line 479) and added a new Discussion section '**Importin α 3 flexibility mediates NF- κ B specificity**' (line 627).

4. The movement described above also appears to have been reported elsewhere for different NLS-cargo that are preferentially recognised by importin α 3 (as reported by the authors in the discussion paragraph beginning at line 580).

Yes, the reviewer is correct that several recent studies have suggested importin α 3 can undergo conformational changes to accommodate topologically complexes NLS-cargos. It is worth noting that all prior works described observed conformational changes either at the N- or C-terminal NLS-binding pockets. This study reveals the first documented case of importin α 3 undergoing conformational changes at both N- and C-terminal NLS-binding pockets to accommodate the p50 and p65 NLSs, respectively. This point is better emphasized in the Discussion (line 627)

5. Please clarify line 237, with regards to the 'docking' of the p50-NLS peptide. 'The p65-NLS was readily modeled in an unbiased Fo-Fc density, while the density for the p50-NLS could be DOCKED at the major NLS-pocket using Coot'.

Sorry, we misspoke here. We meant 'built' in the electron density difference map. The text was revised, see line 253.

6. When RMSD values are reported, please note what residues were aligned in the analysis. For example, in line 360 it is reported that in the two complexes the p50-NLS were superimposable (RMSD 1.095). Were only residues that could be confidently modelled into the electron density compared (see point 6 below)?

Done. The text at line 501 and below as well as **Supplementary Figure 6** have been revised to include the residues used for RMSD analysis and RMSD values.

(Minor points) 1. Figure 2E, Supplemental Figure 2 – how many replicates?

We have repeated these experiments dozens of times, as we extensively tried to crystallize these complexes and kept purifying them for nearly five years.

2. Please define the residues that constitute Δ IBB-importin α 3 and Δ IBB-importin α 1.

Done. See line 142 in the Methods.

3. In Figure 1A please number the regions spanning each domain.

Done. See revised Figure 1.

4. Figure 2D lacks the sizes of the MW marker and therefore one can't correlate the sizes of Δ IBB- α 3 and Δ DBD-p50 between this gel and that in Supplementary Fig 1A and B.

Done. We updated Fig. 2D and labeled the marker.

On this note, I would consider it essential to note the expected MWs of all of the complexes in a table in order to fully evaluate the data.

Done. We added **Supplementary Table 1** that reports the MW of all NF- κ B:Importin α complexes assembled in this study. We refer to this new table in section '**Complex Assembly**', in the methods (see line 201).

5. Δ NLSp50 is reported as residues 41-351. Why are the first 40 residues also missing?

Yes, the original construct of the Δ NLS-p50 was a gift from Dr. Steven Harrison, which was cloned into a pLM1 vector and crystallized with I κ B α . The truncated Δ NLS-p50 construct is identical to the co-crystal structure of p50/p65 with DNA in Chen et al. 1998 and was involved in crystallographic efforts. The first 40 amino acids were found to be cleaved by chymotrypsin in a limited proteolysis assay.

6. Line 350 it is reported that 'All crystal structures had clear electron densities for the p50- and p65-NLSs that we modeled in unbiased F0-Fc electron density maps and confirmed in Polder maps'. In the true definition of the term, both NLS's are entirely visible, but the NLS is defined beyond these key residues in this study (ie p50-NLS is defined as 355-368), and these 'flanking residues' are not visible in the electron density portrayed in Fig 3D and supp. Fig 3A.

Yes. The text was revised to include "NLS and adjacent flanking regions" (line 403).

REVIEWERS' COMMENTS

Reviewer #2 (Remarks to the Author):

This is a major revision of the original manuscript that addresses many criticisms raised by all three reviewers. A wide range of *in vitro* biochemical as well as diverse structural approaches are employed in support of the study. The result is a very nicely performed and communicated report that makes a compelling case for a *trans* bipartite nuclear localization signal that is unique to the NF- κ B p50:p65 heterodimer and recognized specifically by the importin- α 3 nuclear import receptor. These findings are significant and merit publication.

Suggested edit:

Figure 5, Panel G: two mentions of "p65-NLS" incorrectly contain a capital letter "P".

Reviewer #3 (Remarks to the Author):

All of my points of concern have been well addressed. However, ideally ALL of the information conveyed in point 2 of the rebuttal should be included in the revised manuscript for the benefit of the reader.

Point-by-point Response to Reviewers' comments (II round)

We thank the reviewers and the editor for the valuable comments.

Reviewer #2

Figure 5, Panel G: two mentions of "p65-NLS" incorrectly contain a capital letter "P".

Thank you! We changed it to a lower case 'p'

Reviewer #3

All of my points of concern have been well addressed. However, ideally ALL of the information conveyed in point 2 of the rebuttal should be included in the revised manuscript for the benefit of the reader.

Thank you! We have expanded the Figure 5 legend on page 28, line 1075, to include the following text: "(b) The two chains in the asymmetric unit are nearly identical (RMSD = 1.14 Å), but the dimer is likely a crystallographic artifact due to the high concentration used for crystallization. Both importin α 3 molecules in the asymmetric unit are bound to the p50- and p65-NLS peptides, which occupy identical binding sites and have similar structures. However, one of the importin α 3 molecules (chain A) has higher B-factor, likely due to weaker crystal contacts, which prompted us to use chain C for structural analysis".